# Lactoferricins impair the cytosolic membrane of *Escherichia coli* within a few seconds and accumulate inside the cell

Enrico F Semeraro[1,2,3]*, Lisa Marx[1,2,3], Johannes Mandl[1,2,3], Ilse Letofsky-Papst[4], Claudia Mayrhofer[5], Moritz PK Frewein[1,2,3,6], Haden L Scott[7,8], Sylvain Prévost[6], Helmut Bergler[1,2,3], Karl Lohner[1,2,3], Georg Pabst[1,2,3]*

[1]University of Graz, Institute of Molecular Biosciences, NAWI Graz, Graz, Austria; [2]BioTechMed Graz, Graz, Austria; [3]Field of Excellence BioHealth – University of Graz, Graz, Austria; [4]Institute of Electron Microscopy and Nanoanalysis and Center for Electron Microscopy, Graz University of Technology, NAWI Graz, Graz, Austria; [5]Center for Electron Microscopy, Graz, Austria; [6]Institut Laue-Langevin, Grenoble, France; [7]Center for Environmental Biotechnology, University of Tennessee, Knoxville, United States; [8]Shull Wollan Center, Oak Ridge National Laboratory, Oak Ridge, United States

**\*For correspondence:**
enrico.semeraro@uni-graz.at (EFS);
georg.pabst@uni-graz.at (GP)

**Competing interest:** The authors declare that no competing interests exist.

**Abstract** We report the real-time response of *Escherichia coli* to lactoferricin-derived antimicrobial peptides (AMPs) on length scales bridging microscopic cell sizes to nanoscopic lipid packing using millisecond time-resolved synchrotron small-angle X-ray scattering. Coupling a multiscale scattering data analysis to biophysical assays for peptide partitioning revealed that the AMPs rapidly permeabilize the cytosolic membrane within less than 3 s—much faster than previously considered. Final intracellular AMP concentrations of ~80–100 mM suggest an efficient obstruction of physiologically important processes as the primary cause of bacterial killing. On the other hand, damage of the cell envelope and leakage occurred also at sublethal peptide concentrations, thus emerging as a collateral effect of AMP activity that does not kill the bacteria. This implies that the impairment of the membrane barrier is a necessary but not sufficient condition for microbial killing by lactoferricins. The most efficient AMP studied exceeds others in both speed of permeabilizing membranes and lowest intracellular peptide concentration needed to inhibit bacterial growth.

## Editor's evaluation

This article presents groundbreaking data on the effects of a family of antimicrobial peptides on bacterial cells, obtained by time-resolved small-angle X-ray and neutron scattering experiments coupled to stopped-flow mixing. Application of this approach to cells is highly innovative. The main result is that the peptides reach the cytosol in a few seconds, where their accumulation at high concentrations finally kills the bacteria.

## Introduction

Progress in designing antibiotics with novel key-lock mechanisms is not keeping pace with the worldwide growing number of (multi) resistant bacterial strains, encouraging significant research efforts in promising alternatives such as antimicrobial peptides (AMPs) (*Lohner, 2001*). AMPs are part of the

natural innate immune system and provide a first line of defense against pathogens. Their advantage compared to conventional antibiotics relies on a rapid impairment of the barrier function of the bacterial envelope by unspecific physical interactions, often coupled to an ensuing targeting of bacterial DNA or ribosomes (for review, see, e.g., *Wimley and Hristova, 2011*; *Lohner, 2017*; *Malanovic et al., 2020*; *Stella et al., 2021*).

Membrane-active AMPs contain specific sequences of cationic and apolar amino acids, granting high affinity to the hydrophobic core of lipid membranes and selectivity toward the negatively charged surfaces of bacterial envelopes. However, despite intense research for several decades, a comprehensive understanding of the specific series of events that pertain to the bactericidal or bacteriostatic activity of AMPs is still elusive. To a large extent, this is due to the persisting challenge of merging results from in vitro studies with those obtained from lipid membrane mimics, often leading to significant controversies (*Wimley and Hristova, 2011*). This is nurtured, on the one hand, by difficulties in engineering lipid model systems of sufficiently high complexity to mimic the diverse physicochemical properties of bacterial membranes. On the other hand, the complexity of live bacteria challenges experimental and computational techniques to obtain quantitative results on the molecular level. For example, cryogenic transmission electron microscopy (TEM) provides high subcellular spatial resolution, but might give misleading information due to artifacts that potentially originate from staining or invasive sample preparation. Moreover, structural kinetics occurring in the seconds time scale are yet not accessible to cryo-TEM on cells, but would be needed to unravel the sequence of events induced by AMP activity. Kinetic experiments, using high-speed atomic force microscopy, for example, showed an AMP-induced corrugation of the bacterial outer surface within the first 13 s after addition of the AMP, but were limited by the intrinsic time resolution of the technique (*Fantner et al., 2010*). Additionally, such topological experiments do not provide insight into concurring intracellular changes. Video microscopy, combined with fluorescence labeling schemes for peptides or cellular content, in turn provided the appropriate spatiotemporal resolution to differentiate AMP activity in different cells within several tens of seconds (see *Choi et al., 2016* for review). Importantly, such experiments reported that AMPs often reach the cytoplasm within a few minutes, suggesting that the final target for arresting bacterial growth or killing is not the cytoplasmic membrane (*Sochacki et al., 2011*). However, fluorescence labeling may easily tweak the delicate balance of macromolecular interactions and thus affect experimental observations.

Elastic X-ray and neutron scattering experiments are well-established noninvasive techniques to interrogate the structural properties of biological or biobased matter without the need to resort to any bulky labels. However, quantitative analysis of complex biologically relevant systems is challenging and has hampered progress in applying these techniques to live cells (*Semeraro et al., 2021a*). To this end, a full analysis of cells necessitates to account for structural features at diverse hierarchical levels. We recently refined an analytical multiscale scattering model of live *Escherichia coli*, making extensive use of the different sensitivities of X-rays and neutrons to matter, including H/D contrast variation (*Semeraro et al., 2021b*). This allowed us to detail the bacterial hierarchical structure on length scales bridging four orders of magnitude, that is, spanning from bacterial size to the molecular packing of lipopolysaccharides (LPS) in the outer leaflet of the outer membrane. Here, we use this model, taking advantage of the fact that the full breadth of structural information is encoded in a single scattering pattern, and exploit millisecond time-resolved synchrotron (ultra) small-angle X-ray scattering (USAXS/SAXS) to study the response of *E. coli* to three well-characterized lactoferricin-derived AMPs: LF11-215 (FWRIRIRR-NH$_2$), LF11-324 (PFFWRIRIRR-NH$_2$), and O-LF11-215 (octanoyl-FWRIRIRR-NH$_2$) (*Zweytick et al., 2011*; *Zweytick et al., 2014*; *Sánchez-Gómez et al., 2015*; *Marx et al., 2021b*).

Joining these elastic scattering experiments with TEM and assays for determining peptide partitioning as a function of peptide activity enabled us to gain unprecedented insight into the peptide-induced sequence of events. We observed that the studied peptides are able to cause a loss of cytoplasmic content just within a few seconds. Coupling this finding to the derived final (after 1 hr) intracellular peptide concentrations at full bacterial growth inhibition (~80-100 mM) suggests a peptide uptake on similar time scales, and hence much faster than previously reported for other AMPs, such as rhodamine-labeled LL-37 (*Sochacki et al., 2011*). The most effective AMP presently studied, LF11-324, excelled others by most swiftly permeabilizing the cytoplamic membrane and killing the bacteria at the lowest cytosolic/periplasmic concentration. Consistent with previous studies on these AMPs (*Zweytick et al., 2011*), we also observed severe damage of the bacterial cell envelope—here defined

in terms of loss of LPS packing, loss of positional correlations between outer and inner membranes and vesiculation/tubulation. Interestingly, this structural impairment of the bacterial envelope also occurred at peptide concentrations far below the minimum inhibitory concentration (MIC), but on different (slower) time scales than leakage. Leakage kinetics increased with peptide concentration up to the MIC. Yet, the overall loss of cytoplasmic molecules was not affected by AMP concentration, that is, it was equal at MIC and below (even at 1% growth inhibition!). The primary cause of bactericidal or bacteriostatic activity of the presently studied peptides is thus not a damage of the structural integrity of the cell wall, but appears to be due to a fast and efficient impairment of a series of here not further detailed physiological processes occurring within the intracellular space.

## Results

### Defining structural reference states of AMP activity in *E. coli*

Unraveling the timeline of structural events occurring in *E. coli* due to lactoferricin activity by USAXS/SAXS necessitates a detailed prior characterization of two reference states: (i) neat bacteria before AMP administration ('initial state') and (ii) AMP-affected/killed bacteria ('end state'). Here, end state refers to 1 hr of incubation of bacteria at a given AMP concentration. Both states can be treated as *quasi* equilibrium structures, enabling a detailed characterization through a joint application of X-ray and neutron scattering. This allowed us to constrain parameters in the analysis of the time-resolved USAXS/SAXS data described further below. We have recently reported initial-state structures of *E. coli* ATCC 25922 (and other strains) at different hierarchical length scales—including size of bacteria, distance between inner and outer membranes, and LPS packing density—in terms of a multiscale analytical model using joint USAXS/SAXS and (very) small-angle neutron scattering (VSANS)/SANS experiments (*Semeraro et al., 2021b*). Importantly, while this analysis revealed a sensitivity of X-rays/neutrons to structural features of the cellular envelope or bacteria size, we also found that neither flagella or fimbriae, nor macromolecules located in the cytosol (DNA, ribosomes, proteins, etc.) contribute discernible scattering patterns and hence can be detected.

The same methodological concept was applied here to reveal the end-state structure of *E. coli*. Further, and analogously to *Semeraro et al., 2021b*, we coupled the USAXS/SAXS and SANS analysis to TEM in order to remove ambiguities in adjustable parameters originating from ensemble averaging (*Figure 1*). In order to couple scattering experiments to AMP susceptibility assays, it is important to scale peptide concentrations appropriately. Firstly, because SAXS/SANS experiments require up to $10^4$ times higher bacterial concentrations than standard growth-inhibition experiments. Secondly, because the total number of AMPs partitioning into bacteria (and hence also their antimicrobial activity) depends on cell concentration in a nontrivial manner (*Marx et al., 2021b*). We consequently determined the MICs of all here-studied AMPs as a function of cell number density prior to scattering experiments; the corresponding data are reported in *Figure 1—figure supplement 1*.

Small-angle scattering (SAS) patterns of initial and end states showed distinct differences, which are highlighted in *Figure 1—figure supplement 2E and F*. *Figure 1—figure supplement 2E*, in particular, demonstrates a decrease of intensity at very low $q$-values (as approximation of forward scattering), and a faster intensity decay at high $q$ for the end state, coupled to a loss of two intensity wiggles at $q \sim 0.1\,\mathrm{nm}^{-1}$ and $q \sim 0.3\,\mathrm{nm}^{-1}$ (see also *Figure 1A*). The comparison of SANS data at 90% $D_2O$ (*Figure 1—figure supplement 2F*) instead shows a smoothening of an intensity shoulder at $q \sim 0.13\,\mathrm{nm}^{-1}$ coupled to a slight peak-shift toward lower $q$-values for the end state. The application of our previously developed analytical model for *E. coli* (*Semeraro et al., 2021b*) allowed us to gain detailed insight into the ultrastructural changes pertaining to these differences in scattered intensities. To achieve this goal, we jointly analyzed USAXS/SAXS and SANS data at five different contrasts (10, 30, 40, 50, and 90% $D_2O$; see *Figure 1—figure supplement 2*). Taking AMP-induced changes of bacterial ultrastructure as observed by TEM into account then leads to the following observations.

Membrane ruffling, mainly originating from increased fluctuations of cytoplasmic membranes clearly seen by TEM, can be modeled by changing the average distance between inner and outer membranes and its fluctuation (*Figure 1B*), which readily accounts for the above-described change of the SANS feature in *Figure 1—figure supplement 2F*. Consistent with previous reports (*Sochacki et al., 2011*), we also observed an overall shrinking of bacterial size. This correlates with a loss of

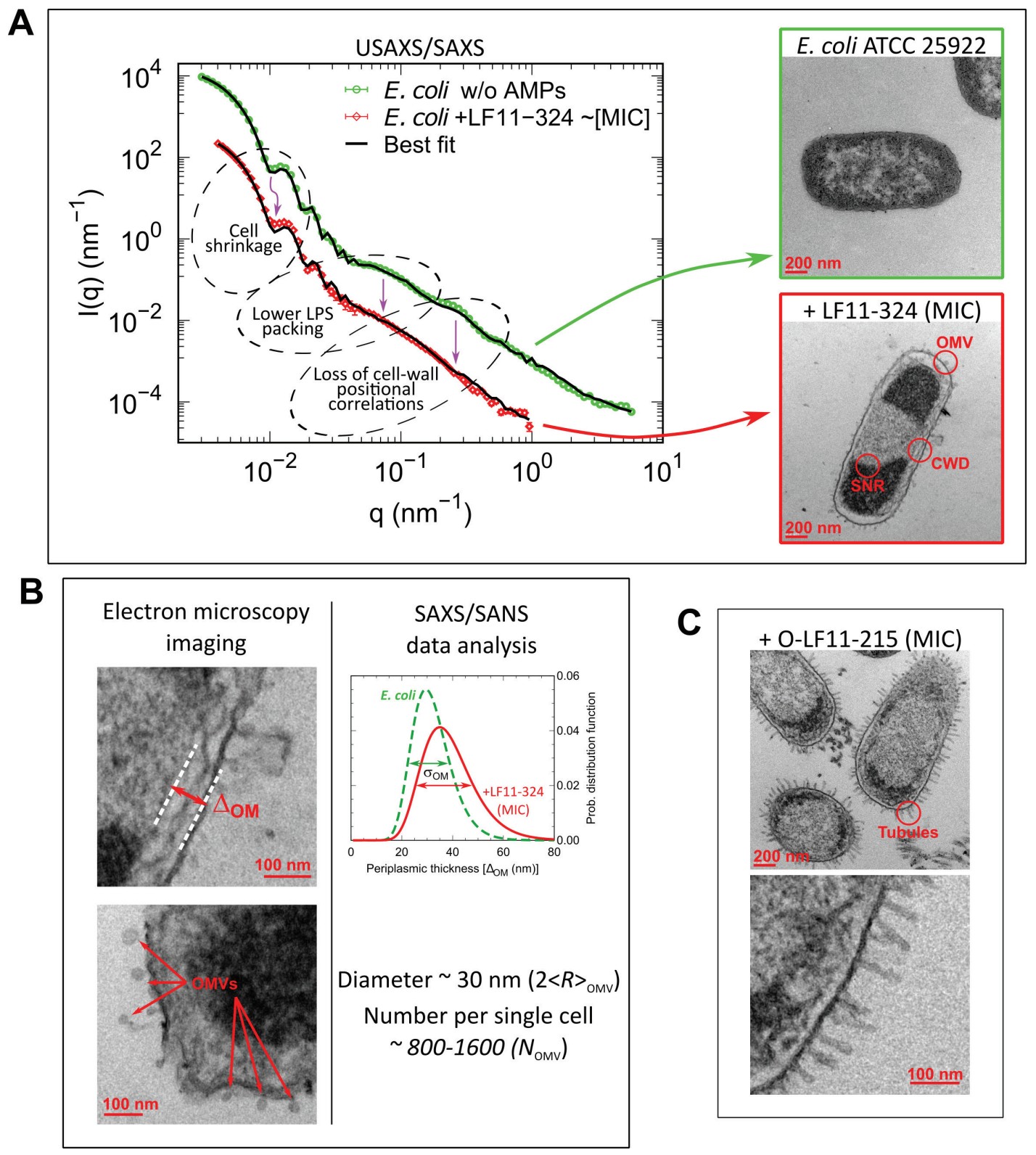

**Figure 1.** Overview of combined X-ray scattering and electron-microscopy measurements. (**A**) Mapping the main structural changes in *E. coli* ATCC 25922 (green symbols) upon 1 hr incubation with LF11-324 (red symbols) as observed by (ultra) small-angle X-ray scattering (USAXS/SAXS) and transmission electron microscopy (TEM). Scattering data of *E. coli* ATCC 25922 are from *Semeraro et al., 2021b* and have been obtained at 10-fold higher sample concentration, leading to the observed offset of scattered intensities. Black lines are the best fits using *Equation 6*. OMV: outer

*Figure 1 continued on next page*

*Figure 1 continued*

membrane vesicle formation; CWD: cell-wall damaging; SNR: phase separation of the nucleoid region. Error bars are given by the experimental error of the measurements. (**B**) TEM examples of membrane detachment and OMV formation due to LF11-324, and respective *ensemble* results from scattering data analysis for the distance distribution between inner and outer membranes. (**C**) Bacteria upon 1 hr incubation with O-LF11-215, showing the formation of tube-like protrusions.

The online version of this article includes the following figure supplement(s) for figure 1:

**Figure supplement 1.** Cell number-dependent minimum inhibitory concentration (MIC) plots for different peptides.

**Figure supplement 2.** Comparison between (ultra) small-angle X-ray scattering (USAXS/SAXS) and contrast-variation small-angle neutron scattering (SANS) data and details of the scattering data analysis.

**Figure supplement 3.** Transmission electron microscopy (TEM) observations for LF11-215, LF11-324, and O-LF11-215 at the minimum inhibitory concentrations (MICs) and sub-MICs.

**Figure supplement 4.** Graphical scheme of the adjustable parameters.

periplasmic or cytosolic material. Both effects explain the changes in USAXS intensity at very low $q$-values (*Figure 1—figure supplement 2E*).

The scattering shoulder in USAXS/SAXS data at $q \sim 0.07\,\mathrm{nm}^{-1}$ originates from positional correlations between LPS oligosaccharide cores, as revealed by our previous in-depth analysis (*Semeraro et al., 2021b*). Its smearing out in the presence of peptide consequently indicates a loss of LPS packing correlations, either because of a decrease of the number of LPS molecules on the outer surface or due to increased membrane roughness or waviness. This effect occurs on sub-nanometer length scales and is thus hidden from TEM. Notably, the end-state scattering patterns in the presence of either LF11-324 or LF11-215 were identical, despite their distinct MIC values (data not shown). Modeling their scattering data revealed an additional contribution, which we could associate with the help of TEM to the formation of outer membrane vesicles (OMVs), having an average diameter of ~30 nm (*Figure 1A and B*, *Figure 1—figure supplement 2E*). Instead, the separation of the nucleotide region from the nucleotide-free cytosol revealed by TEM (*Figure 1A*), and in agreement with a previous report on same AMPs (*Zweytick et al., 2011*), is not observed in our scattering data. As noted above, SAS is not sensitive to macromolecules and aggregates thereof inside the cell (*Semeraro et al., 2021b*). Finally, O-LF11-215 led to additional effects. First of all, O-LF11-215, because of its increased hydrophobicity, forms aggregates in buffer as clearly shown by SAXS (Appendix 1). In particular, incubating O-LF11-215 with bacteria at concentrations needed for SAS experiments led to the formation of macroscopic aggregates, which impeded the measurements of end states by SAS. TEM experiments, however, showed the formation of extramembranous tubes (*Figure 1C*).

**Table 1.** Change of *E. coli* structure due to LF11-324 ([P] $\sim$ MIC) as observed from USAXS/SAXS/SANS data analysis.
Values are the difference between end- and initial state. See a graphical scheme of the adjustable parameters in *Figure 1—figure supplement 4*, and *Semeraro et al., 2021b* for a more detailed schematic.

| Parameters | Values | Description |
|---|---|---|
| $\Delta\rho_{\mathrm{CP}} \times 10^{-4}$ (nm$^{-2}$) | −0.17 ± 0.02*; −0.14 ± 0.05† | $\rho_{\mathrm{CP}} \to$ SLD of cytoplasmic space |
| $\Delta\rho_{\mathrm{PP}} \times 10^{-4}$ (nm$^{-2}$) | 0.18 ± 0.06*; 0.12 ± 0.04† | $\rho_{\mathrm{PP}} \to$ SLD of periplasmic space |
| $\Delta\Delta_{\mathrm{OM}}$(nm) | 6 ± 3 | $\Delta_{\mathrm{OM}} \to$ inter-membranes distance |
| $\Delta\sigma_{\mathrm{OM}}$(nm) | 3.4 ± 1.7 | $\sigma_{\mathrm{OM}} \to$ SD around $\Delta_{\mathrm{OM}}$ |
| $\Delta\rho_{\mathrm{PG}} \times 10^{-4}$ (nm$^{-2}$) | −0.27 ± 0.07*; −0.59 ± 0.14† | $\rho_{\mathrm{PG}} \to$ SLD of peptidoglycan layer |
| $\Delta p_{\mathrm{LPS}}$ | −0.44 ± 0.08*; −0.26 ± 0.11† | $p_{\mathrm{LPS}} \to$ LPS packing parameter |
| $\Delta R$(nm) | −27 ± 7 | $R \to$ minor radius of the cytoplasmic space |

Differences in X-ray and neutron SLDs are due to different physical interactions with matter. In case of $p_{\mathrm{LPS}}$, this originates from a biological variation of different bacterial cultures.
USAXS/SAXS: (ultra) small-angle X-ray scattering; SANS: small-angle neutron scattering; SLD: scattering length density; LPS: lipopolysaccharide.
*From SAXS.
†From SANS (SLDs were obtained by extrapolating to 0 wt% D$_2$O); see *Figure 1—figure supplement 2B–D*.

All end-state SAS data were fitted with our analytical model using a Monte Carlo genetic-selection algorithm; see *Figure 1—figure supplement 4* for a graphical scheme of all adjustable parameters. Compared to standard least-square algorithms, this allowed us to obtain robust parameter values from probability density functions and to judge possible parameter correlations (for details, see 'Materials and methods'). Based on our previous modeling of neat *E. coli* (*Semeraro et al., 2021b*), the individual SANS contrasts (*Figure 1—figure supplement 2A*) are motivated as follows: (i) the distance between the inner and outer membranes $\Delta_{OM}$ and its fluctuations, $\sigma_{OM}$, is accentuated at 90, 50, and 40% $D_2O$; (ii) 30–50% $D_2O$ highlight the scattering length density (SLD) balance between the cytosol $\rho_{CP}$, periplasm $\rho_{PP}$, and the peptidoglycan layer $\rho_{PG}$; and (iii) 10% $D_2O$ yields a similar contrast than X-ray and thus provides a direct and independent control for USAXS/SAXS experiments. *Table 1* summarizes the results of this analysis for LF11-324 by listing the changes between initial and end states. As discussed above, our analysis also reports overall bacterial size changes. *E. coli* are reasonably well represented in reciprocal space using an ellipsoid with multiple shells to account for the different compartments (*Semeraro et al., 2021b*). Since our smallest probed scattering vector, $q_{min} \sim 3 \times 10^{-3}$ nm$^{-1}$ (corresponding to distances of about 2 µm), does not allow to probe the full length of the bacteria, we report size changes in terms of changes of the minor radius of the ellipsoid and, in particular, that of the cytoplasmic space $R$. The major radius was coupled to $R$ by fixing the major to minor radius ratio, $\varepsilon$, to a value of 2 (see Table 2). Finally, the USAXS/SAXS scattering shoulder at $q \sim 0.07$ nm$^{-1}$ reports on positional correlations between LPS oligosaccharide cores (*Semeraro et al., 2021b*). In our analytical model, this is captured by the number of oligosaccharide cores (OS), $N_{OS}$. To characterize the peptide-induced changes, we define the LPS packing parameter, $p_{LPS} = N_{OS}/N_{OS}^0$, as the ratio between the number of OS (i.e., LPS) in the current and initial state ($N_{OS}^0$). Other parameters of the analytical model, such as the thicknesses of the hydrophobic and hydrophilic layers of the cytoplasmic and outer membranes, as well as their average SLDs, the thickness of the peptidoglycan layer, and the radius of gyration of the oligosaccharides, were fixed at the same values as reported in *Semeraro et al., 2021b* (see also Tables 2 and 3). We additionally checked for contributions of peptides to the SLDs of each bacterial compartment, including cytoplasmic (CM) and outer membranes (OM). However, the associated changes were found to be insignificant.

The observed LF11-324-induced changes to *E. coli* show, upon the application of our model, a decrease of $\rho_{CP}$, along with the increase of $\rho_{PP}$, signifying leakage of mainly low-weight molecules and ions from the cytoplasm (*Figure 1—figure supplement 2B and C*), whose scattering dominate both $\rho_{CP}$ and $\rho_{PP}$ (*Semeraro et al., 2021b*). Surprisingly, final $\rho_{CP}$ and $\rho_{PP}$ values did not depend on peptide concentration, that is, were equal within experimental uncertainty at MIC and sub-MICs (see *Figure 2B*). The overall increase of the distance between inner and outer membranes, $\Delta_{OM}$, corresponds to about 18%; its fluctuations $\sigma_{OM}$ increase by almost a factor of two. The associated disorder in the periplasm then also accounts for the drop of the contrast of the peptidoglycan layer $\rho_{PG}$ by about 2.6%. The LPS packing parameter, $p_{LPS}$, instead drops dramatically by 30–40%. Finally, the observed cell shrinkage amounts to ~5%, which leads to a decrease of cell surface of approximately 2 $\times 10^6$ nm$^2$. Apparently this is at least in part compensated by OMV formation; indeed the overall cell-surface decrease is on the same order of magnitude as the estimate of the total surface of all OMVs of $\sim (2-6) \times 10^6$ nm$^2$ as obtained from analyzing SAS data (see Appendix 2).

## Kinetics: Time-resolved USAXS/SAXS

The structural transitions from initial to end state were followed by USAXS/SAXS at millisecond time resolution. Stopped-flow mixing ensured thorough and rapid re-dispersion (mixing time of 50 ms) of peptides and bacteria (*Figure 2—figure supplement 1*) and led to immediate changes of scattering patterns. Using the initial and end-state values obtained for the adjustable parameters as constraints allowed us to fit kinetic scattering data as detailed in the 'Materials and methods'. A close examination of the results obtained for the three LF11s (*Figure 2*, *Figure 2—figure supplement 2*, and *Figure 2—figure supplement 3*) allowed to discern AMP concentration-independent parameters ($p_{LPS}$, $\Delta_{OM}$, $\sigma_{OM}$, and $\rho_{PG}$) from AMP concentration-dependent parameters ($\rho_{CP}$, $\rho_{PP}$, and $R$).

First, we focus on LF11-324 and AMP concentration-independent parameters. The packing of LPS started to decrease at $\Delta t \sim 10$ s after mixing (*Figure 2A*). Changes of $\Delta_{OM}$, $\sigma_{OM}$, and $\rho_{PG}$ in turn are largely decoupled from this remodeling of the outer membrane, with a common onset of 2–10 min after peptide addition (*Figure 2D–F*). AMP concentration-dependent parameters instead showed an

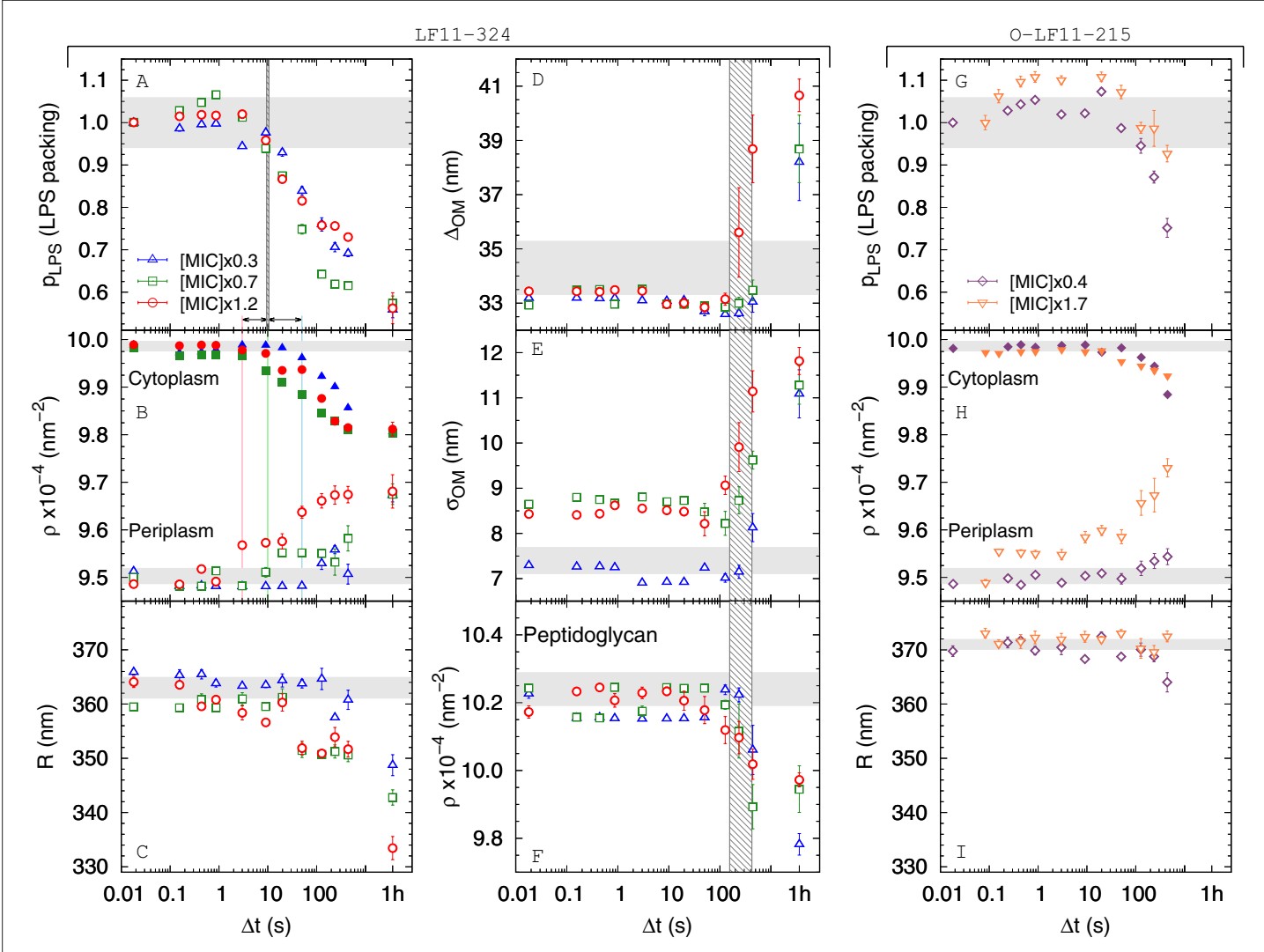

**Figure 2.** Kinetics of the bacterial structural response upon addition of peptide. (**A–F**) Kinetics of the bacterial structural response to attack by LF11-324; results for three different peptide concentrations are shown. Lipopolysaccharide (LPS) packing (**A**); cytoplasm and periplasm scattering length density (SLD) (**B**); minor radius of the cell (**C**); intermembrane distance (~ periplasm thickness) (**D**) and its deviation (**E**); and peptidoglycan SLD (**F**). (**G–I**) Bacterial response to O-LF11-215 at two concentrations. LPS packing (**G**); cytoplasm and periplasm SLDs (**H**); and minor ellipsoidal radius of the cell (**I**). Thick gray bands mark the degree of confidence from bacterial systems w/o peptides (see **Table 1** and **Semeraro et al., 2021b**), except for (**C**) and (**I**), where they refer to the average of the current cell radii at $\Delta t = 0.0175\,\text{s}$. Fluctuations of initial values can be due to biological diversity. The vertical gray grid (**A, D–F**) indicates the time range of local (**A**) and macroscopic (**D–F**) cell-wall damage. Note that this range does not depend on peptide concentration. Colored lines in (**B**) mark the *concentration-dependent* lower boundary for the onsets of leakage. Results at $\Delta t = 1\,\text{hr}$ refer to end states, when available. Error bars are given by the associated standard deviations of the adjustable parameters obtained from the analysis of scattering curves.

The online version of this article includes the following figure supplement(s) for figure 2:

**Figure supplement 1.** Schematic of the stopped-flow rapid mixing (ultra) small-angle X-ray scattering (USAXS/SAXS) experiments, including selected scattering patterns.

**Figure supplement 2.** Kinetics of the adjustable parameters for LF11-215 and O-LF11-215 systems.

**Figure supplement 3.** Kinetics of the adjustable parameters for LF11-215 systems.

**Figure supplement 4.** Representative scattering patterns including errors and fitted curves.

**Figure supplement 5.** Representative distributions of the adjustable parameters at different time points.

**Figure supplement 6.** Representative series of correlation plots and coefficients of the adjustable parameters.

increasing delay of changes upon decreasing the amount of administered peptide (*Figure 2B and C*). In particular, $\rho_{PP}$ exhibited a pronounced increase already at $\Delta t \sim 3\,\mathrm{s}$ at highest peptide concentration shifting to $\Delta t \sim 2\,\mathrm{min}$ at the presently lowest studied peptide concentration. The onset times of decrease of $\rho_{CP}$ do not appear to be directly correlated with these changes, but also shifted progressively to later times with decreasing AMPs. Because of the dominant contribution of low-molecular-weight molecules and ions to both $\rho_{CP}$ and $\rho_{PP}$ (*Semeraro et al., 2021b*), we surmise that these changes are due to a leakage of inner and outer membranes. Briefly, the cytoplasmic content diffuses first in the periplasmic space and, simultaneously, material from the periplasm leaks out of the cell. This process leads to the rather simultaneous decrease of $\rho_{CP}$ and increase of $\rho_{PP}$. Because of differences in the individual onsets of these trends at a given AMP concentration, we can only associate a time range for the beginning of permeation of the cytoplasmic membrane: 3–10 s at $[P] = 1.2 \times \mathrm{MIC}$, 10–20 s at $[P] = 0.7 \times \mathrm{MIC}$, and 50–120 s at $[P] = 0.3 \times \mathrm{MIC}$. Despite these differences, and as noted above, $\rho_{CP}$ (and $\rho_{PP}$) reached the *same* final values for all three AMP concentrations. Finally, the drop of $R$ started at 20–50 s for $[P] = 1.2 \times \mathrm{MIC}$ and $0.7 \times \mathrm{MIC}$, and >10 min for $[P] = 0.3 \times \mathrm{MIC}$ (*Figure 2C*).

Interestingly, LF11-215 led to almost identical kinetics for the concentration-independent parameters ($\Delta_{OM}$, $\sigma_{OM}$, and $\rho_{PG}$ in *Figure 2—figure supplement 2A–C* and $p_{LPS}$ in *Figure 2—figure supplement 3A*). This suggests that LF11-215 and LF11-324 remodel the membranes and affect the macroscopic stability of the cell envelope in a very similar fashion, although we found no decrease of $p_{LPS}$ for LF11-215 at $[P] = 1.6 \times \mathrm{MIC}$ (*Figure 2—figure supplement 3A*). Importantly, $\rho_{CP}$ and $\rho_{PP}$ started to change much later for this peptide ($\Delta t \sim 20-50\,\mathrm{s}$ at $1.6 \times \mathrm{MIC}$, *Figure 2—figure supplement 3B*). The overall changes of $\rho_{CP}$ and $\rho_{PP}$ indicate, however, that the AMP concentration-induced loss of cytoplasmic content is a common feature of LF11-324 and LF11-215.

The kinetics of the cell-envelope damage caused by O-LF11-215 were found to proceed analogously to both LF11-324 and LF11-215 (*Figure 2—figure supplement 2D–F*). However, while changes of $\rho_{CP}$ and $\rho_{PP}$ occurred at about similar times than for LF11-324 (*Figure 2B and H*), $p_{LPS}$ ($\Delta t \sim 2\,\mathrm{min}$, *Figure 2G*) and $R$ ($\Delta t \sim 10\,\mathrm{min}$, *Figure 2I*) changed much later. The initial increase of $p_{LPS}$ is likely to be an artifact, probably due to the convolution of scattering originating from cells and peptide aggregates (see *Figure 2—figure supplement 1D*).

LF11-215/LF11-324-induced OMV formation was discernible for $\Delta t > 1–2\,\mathrm{min}$ (*Figure 3A*) and appears to be peptide- and concentration-independent within the first 10 min. Finally, the peptide cluster term, introduced for the analysis of O-LF11-215 (see 'Materials and methods'), enabled us to estimate that a large increase of peptide uptake starts after about 2 min, which, however, did not depend on peptide concentration (*Figure 3B*).

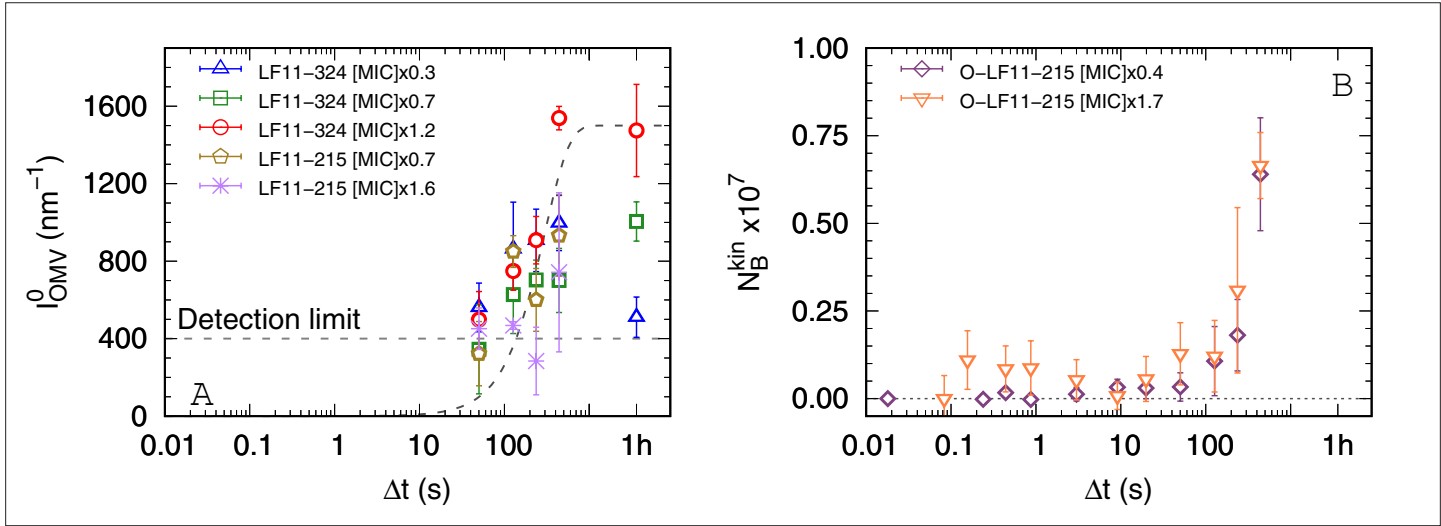

**Figure 3.** Kinetics of outer membrane vesicle formation and O-LF11-215 absorption. (**A**) Kinetics of the forward intensity of outer membrane vesicle (OMV) scattering for different concentrations of LF11-324 and LF11-215. The dashed horizontal line represents the detection/'visibility' limit, below which $I_{\mathrm{cell}}(q) + I_{\mathrm{OMV}}(q) \approx I_{\mathrm{cell}}(q)$ in the entire q-range. The dashed exponential curve is a guide for the eyes. (**B**) Evolution of the number of partitioned peptides per cell for two O-LF11-215 concentrations, as calculated from the analysis of $I_{\mathrm{clu}}$. Error bars are derived from the associated standard deviations of the adjustable parameters obtained from the analysis of scattering curves.

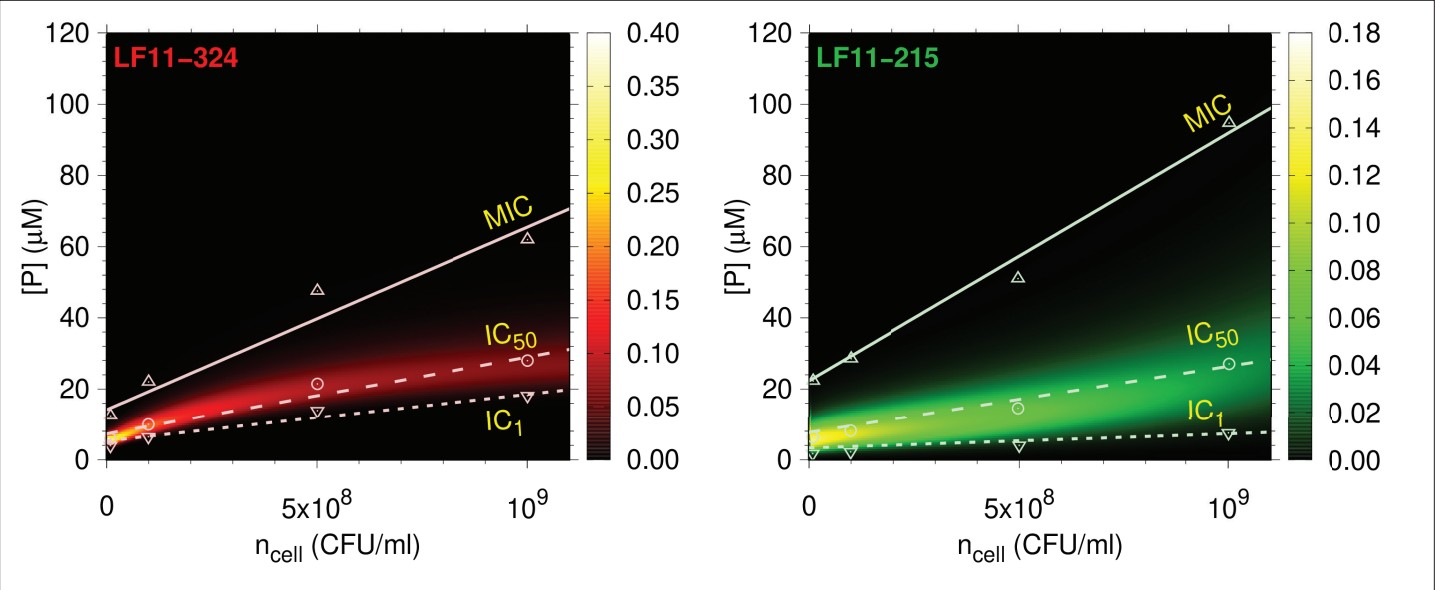

**Figure 4.** Full partitioning maps for LF11-324 and LF11-215. Amount of LF11-324 or LF11-215 required to attain growth-inhibited fractions of either 0.999 (minimum inhibitory concentration [MIC], up triangles), 0.5 (circles), or 0.01 (down triangles) in *E. coli* ATCC 25922 as a function of $n_{cell}$. Lines are fits with *Equation 1*. These data are overlaid with a surface plot of the associated killing probability density function. The color scales indicate the corresponding magnitudes.

The online version of this article includes the following figure supplement(s) for figure 4:

**Figure supplement 1.** Inhibitory concentration (IC$_x$) as a function of $\phi_{IG}$ (inverse cumulative distribution function [CDF]).

## Peptide partitioning and cooperativity

We applied a previously detailed assay for AMP partitioning in *E. coli* based on growth inhibition (*Marx et al., 2021b*). A statistical analysis of the corresponding data in terms of cumulative distribution functions (CDFs, see Appendix 3) allowed us to derive probability density functions (PDFs), describing bacterial growth inhibition as a function of cell concentration $n_{cell}$, including the minimum AMP concentration needed for inhibiting a given percentage $x$ of *E. coli*, IC$_x$ (*Figure 4*); note that $\mathrm{MIC} \equiv \mathrm{IC}_{99.9}$. We obtained Poisson-like PDFs of 'killing' events as a function of peptide concentration from the sigmoidal shapes of the measured CDFs (Appendix 3). Besides the specific mode of action of a peptide, the shape, width, and position of the PDFs are a result of intrinsic statistical variations and fluctuations of the systems' properties, for example, bacterial size, stage of mitosis, local concentration of partitioned peptides, biological variability within the populations, etc.

In agreement with our previous report for *E. coli* K12 (*Marx et al., 2021b*), the MIC of LF11-324 is lower than that of LF11-215 at all cell concentrations. For O-LF11-215, MIC values matched those of LF11-324 at low cell concentrations, but increased strongly with $n_{cell}$, finally superseding that of LF11-215 and becoming immeasurably high due to the above-described peptide aggregation (*Figure 1—figure supplement 1*). The growth inhibition probabilities for LF11-324 and LF11-215 peak close to the IC$_{50}$s (*Figure 4*). The distributions are much sharper, that is, have a smaller full-width-at-half-maximum (FWHM) of the probability distributions, $\sigma_{[P]}$, for LF11-324 than for LF11-215. This suggests an increased 'cooperativity' of killing for LF11-324, in the sense of a two-state transition model between alive and dead bacteria (see Appendix 3). $\sigma_{[P]}$ increased with cell concentration, for example, from $\sigma_{[P]} \simeq 2.6\,\mathrm{M}$ at $n_{cell} = 10^7$ CFU/ml to $\sigma_{[P]} \simeq 13\,\mathrm{M}$ at $n_{cell} = 10^9$ CFU/ml for LF11-324. Significant noise levels in growth inhibition data for O-LF11-215 impeded a determination of killing probabilities at inhibitory concentrations $< \mathrm{IC}_{50}$. However, data retrieved at higher inhibitory concentrations suggest that the probability distributions roughly match those of LF11-324 at low cell concentrations, but become broader than that of LF11-215 at high cell content (*Figure 4—figure supplement 1*). This is a signature of loss of killing efficacy at high $n_{cell}$, most likely due to peptide self-aggregation as discussed above.

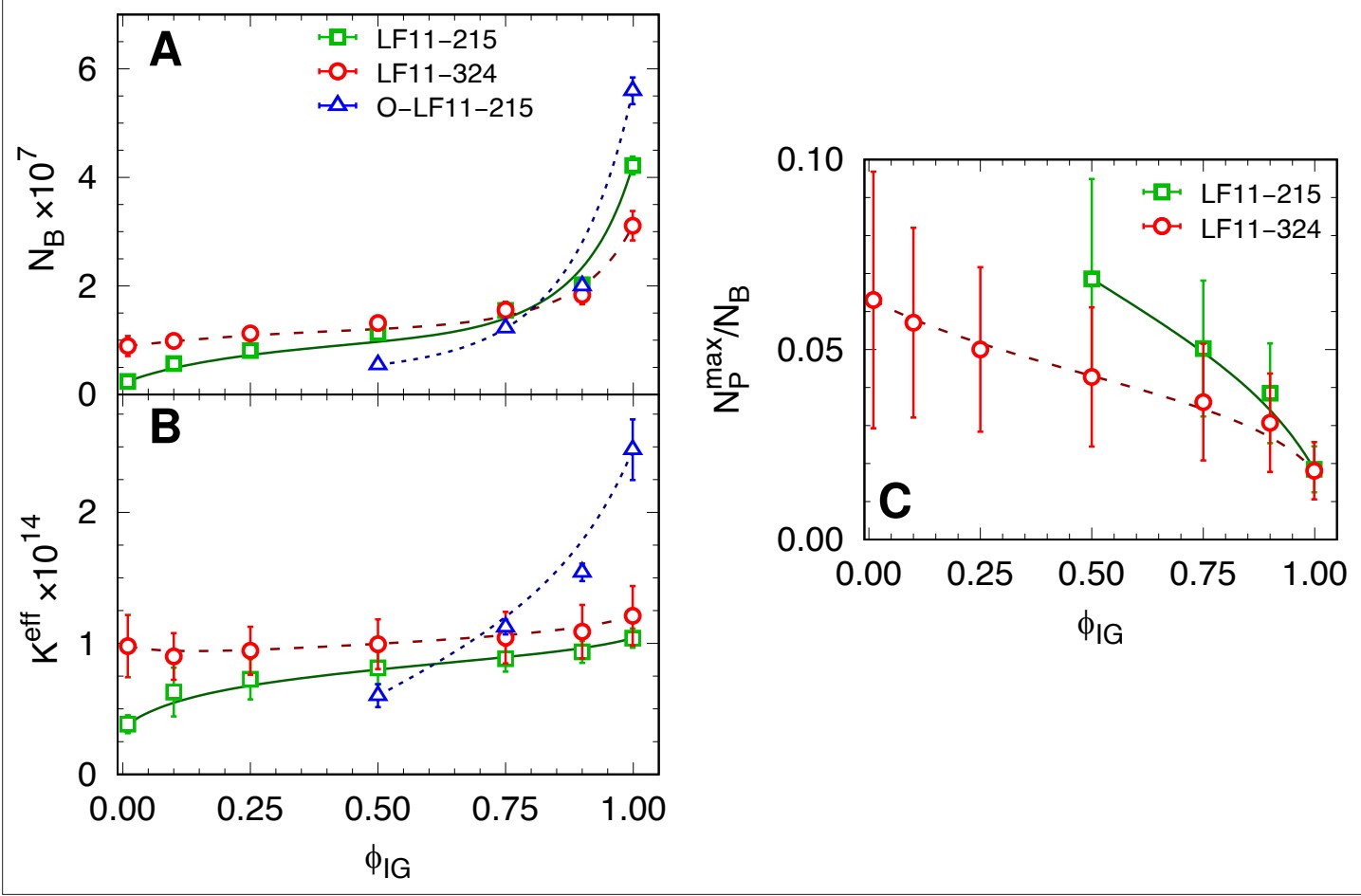

**Figure 5.** Partitioning parameters as a function of cell-growth inhibition. (**A, B**) $N_B$ and $K^{eff}$ values as a function of inhibited fraction. In the case of O-LF11-215, $N_B \equiv N_B^{eff}$. (**C**) Ratio between the maximum number of peptides on the outer leaflet and total number of partitioned peptides, $N_P^{max}/N_B$, as a function of inhibited fraction. Lines are guides for the eye. Error bars in panel (**A** and **B**) are given by the associated standard deviations of the adjustable parameters obtained from the analysis of the equi-activity assay. In panel (**C**) the errors are given by combining errors shown in panel (**A**) with the experimental error propagated from $\zeta$-potential measurements (see *Figure 5—figure supplement 1*).

The online version of this article includes the following figure supplement(s) for figure 5:

**Figure supplement 1.** $\zeta$-potential and size measurements of LF11-215 and LF11-324.

**Figure supplement 2.** $\zeta$-potential and size measurements of O-LF11-215.

Next, we derived for each IC $_x$ (or growth-inhibited fraction, $\phi_{IG}$) the average number of cell-partitioned peptides, $N_B$, and the effective partitioning coefficient, $K^{eff}$, applying a previously reported thermodynamic formalism (*Marx et al., 2021b*; see also *Equation 1*). $N_B$ increased for all three peptides with $\phi_{IG}$, although the changes of $N_B$ were smallest for LF11-324, followed by LF11-215 and O-LF11-215 (*Figure 5A*). These results were confirmed independently also by Trp-fluorescence spectroscopy (see Appendix 4).

This partitioning behavior was also mirrored in the $\phi_{IG}$ dependence of $K^{eff}$. $K^{eff}$ was nearly constant for LF11-324, increased only slightly for LF11-215, and showed the largest variation for O-LF11-215, reaching about 2.5 times higher levels than the other two peptides (*Figure 5B*). The approximate equal $K^{eff}$ values of LF11-324 and LF11-215 for $\phi_{IG} > 0.5$ demonstrate that both peptides partition about equally well into *E. coli*, not only at the MIC, but in a wide range of $\phi_{IG}$ values.

$\zeta$-potential measurements were performed ~5 min and 1 hr (end state) after adding AMPs. Results did not show significant differences for the two time points. This enabled us to further discriminate the activities of LF11-324 and LF11-215. We observed an initial increase of $\zeta/\zeta_0$ ($\zeta_0$ refers to the reference system, i.e., neat bacteria) at low peptide concentrations, reaching plateau values of $\zeta/\zeta_0 = 0.80 \pm 0.16$

for LF11-215 and $\zeta/\zeta_0 = 0.85 \pm 0.17$ for LF11-324 for $[P] \gtrsim 0.3 \times \text{MIC}$ (**Figure 5—figure supplement 1**). As detailed in **Marx et al., 2021b**, these data allowed us to calculate the *maximum* number of peptides associated to the LPS leaflet, $N_P^{max}$. We found $N_P^{max} = (8 \pm 2) \times 10^5$ and $(6 \pm 2) \times 10^5$ for LF11-215 and LF11-324, respectively, in the $\zeta/\zeta_0$-plateau regions. Normalizing these results by the number of cell-partitioned peptides ($N_B$) reveals an overall decrease of $N_P^{max}/N_B$ with $\phi_{\text{IG}}$ (see **Figure 5C**). Note that the $\zeta$-potential analysis provides (with the applied mathematical and physical constrains) upper-boundary values for $N_P^{max}/N_B$ (see 'Materials and methods'). That is, $N_P^{\text{true}}/N_B \leq N_P^{max}/N_B$. Thus, our conservative $N_P^{max}/N_B$ ratio strikingly demonstrates that most of the peptides are located within the intracellular compartments of *E. coli* at the MIC. The fraction of outer-leaflet-partitioned peptides increased toward lower $\phi_{\text{IG}}$, and somewhat stronger for LF11-215, but does not exceed 10%. An analogous analysis for O-LF11-215 was impeded by the peptide aggregates, whose sizes were in the same order or even larger than that of bacteria (**Figure 5—figure supplement 2**).

## Discussion

The entire set of the above-detailed results reveals a complex scenario that needs to be evaluated with utmost care. To this end, we stress that all techniques used in this work report *ensemble* averages. That is, they account for the entire bacterial population within the samples. For example, an LPS packing value of $p_{\text{LPS}} \simeq 0.55$—observed for growth-inhibited fractions of 1%, as well as 99.9% after 1 hr (**Figure 2A**)—signifies that the *ensemble* average shows the same loss of LPS packing under both conditions, irrespective of individual differences in some isolated cells.

A particularly striking result for the LF11-215/LF11-324 *end states* is that the peptide-induced effects are similar and independent of peptide concentration (**Figures 1 and 2**). That is, even at growth-inhibited fractions of just 1%, we observed much the same cellular permeabilization and structural changes of the bacterial ultrastructure as at quasi fully growth-inhibited *E. coli* (see also **Figure 4** and **Figure 4—figure supplement 1**). Here, it is important to bear in mind that, unlike our susceptibility assays, we did not add a growth medium after incubation with peptides in SAS experiments. Our combined SAS and growth inhibition data provide unambiguous evidence that bacteria are able to recover at sub-MIC concentrations from a severe collateral damage of their cell envelope. Consequently, this damage cannot be the primary killing cause of bacteria. Our structural studies only revealed peptide-specific and concentration-dependent effects for the kinetics of $R$, $\rho_{\text{CP}}$, and $\rho_{\text{PP}}$ (**Figure 2**). It is unlikely though that antimicrobial activity is specific to the rate of bacteria size changes. In turn, the different onsets of changes of $\rho_{\text{CP}}$ and $\rho_{\text{PP}}$ may provide some clues to events that finally inhibit the growth of *E. coli*. In what follows, we will thus focus on the sequence of events that can be deduced from our study for LF11-324 and LF11-215. Before doing so, we note that also O-LF11-215 caused comparable variations of the abovementioned parameters on similar time scales. In this case, however, an equally detailed analysis was impeded by the propensity of O-LF11-215 to aggregate in buffer solution.

**Figure 6** provides a scheme of the timeline of events revealed by this study, taking also into account our complementary experiments on peptide partitioning. Within a few seconds, LF11-324 and LF11-215 cause changes of the LPS packing density ($\Delta t \sim 10$ s), as well as periplasmic and cytoplasmic SLDs. Since ions and small molecules (<1 kDa) are the major contributors to its X-ray SLD (**Semeraro et al., 2021b**), the decrease of $\rho_{\text{CP}}$ will be dominated by a leakage of these entities (e.g., ATP). Molecules leaking from the cytosol first diffuse into the periplasm and then further into extracellular space, leading to a net increase of $\rho_{\text{PP}}$. Leaking into extracellular space follows from the observation that final $\rho_{\text{PP}}$ levels do not reach those of $\rho_{\text{CP}}$, despite the much larger reservoir for cytosolic molecules. Further, the initial SLDs of buffer and periplasm are comparable, also explaining why our technique is not directly detecting outer membrane leakage. Hence, either observed change of $\rho_{\text{PP}}$ or $\rho_{\text{CP}}$ is due to a permeabilization of both cytoplasmic and outer membranes.

For LF11-324, the permeabilization of the cytoplasmic membrane occurred as fast as 3–10 s after mixing at $[P] = 1.2 \times \text{MIC}$. Dropping peptide concentration led to a slowing down of this effect (10–20 s for $[P] = 0.7 \times \text{MIC}$, and 50–120 s for $[P] = 0.3 \times \text{MIC}$), but *did not* affect the final cytoplasmic density, that is, the overall loss of material (**Figure 2**). AMPs need to translocate all the way through the cell wall in order to induce such effects, implying that peptide translocation of the outer membrane proceeds on time scales faster than $\Delta\rho_{\text{PP}}$ or $\Delta\rho_{\text{CP}}$. We stress that the usage of the term 'translocate' does not

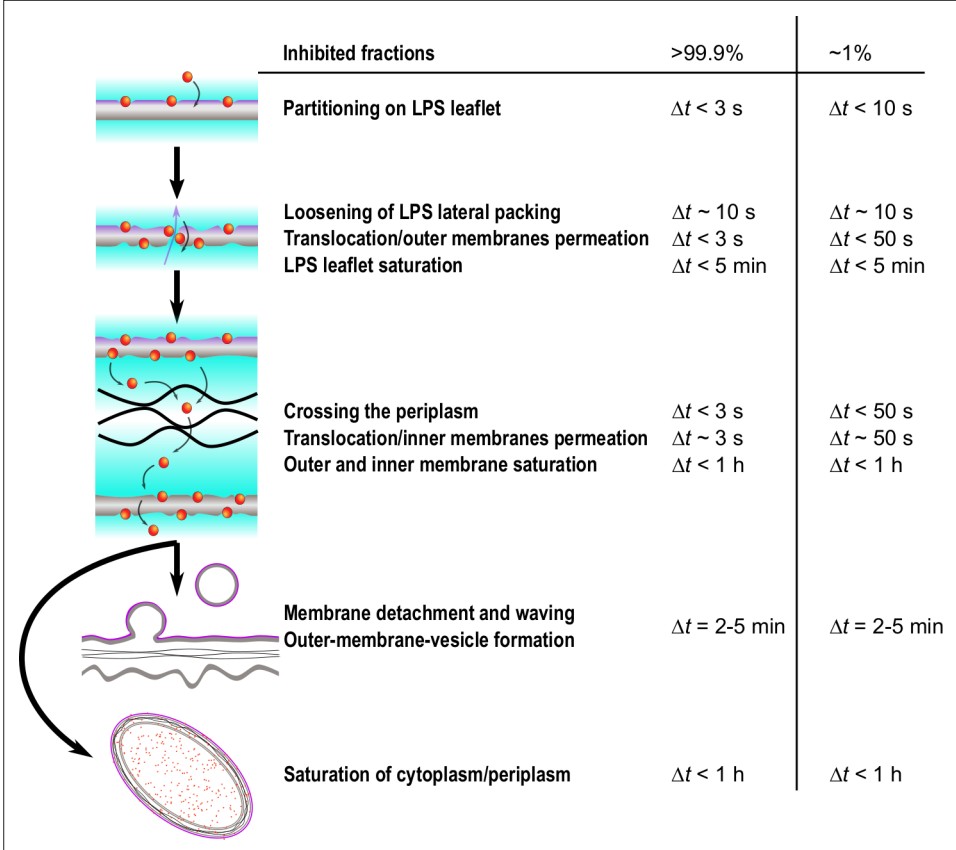

| Inhibited fractions | >99.9% | ~1% |
|---|---|---|
| Partitioning on LPS leaflet | $\Delta t < 3$ s | $\Delta t < 10$ s |
| Loosening of LPS lateral packing | $\Delta t \sim 10$ s | $\Delta t \sim 10$ s |
| Translocation/outer membranes permeation | $\Delta t < 3$ s | $\Delta t < 50$ s |
| LPS leaflet saturation | $\Delta t < 5$ min | $\Delta t < 5$ min |
| Crossing the periplasm | $\Delta t < 3$ s | $\Delta t < 50$ s |
| Translocation/inner membranes permeation | $\Delta t \sim 3$ s | $\Delta t \sim 50$ s |
| Outer and inner membrane saturation | $\Delta t < 1$ h | $\Delta t < 1$ h |
| Membrane detachment and waving Outer-membrane-vesicle formation | $\Delta t = 2\text{-}5$ min | $\Delta t = 2\text{-}5$ min |
| Saturation of cytoplasm/periplasm | $\Delta t < 1$ h | $\Delta t < 1$ h |

**Figure 6.** Simplified time sequence of LF11-215 and LF11-324 mode of action. The measured time onsets and boundaries refers to LF11-324 at minimum inhibitory concentration (MIC) ×0.3 and MIC ×1.2, which correspond to measured inhibited fractions of ~1 and > 99.9%, respectively. The outer leaflet is affected by peptides within the first seconds after their attack. Then, depending on antimicrobial peptide (AMP) type and concentration, a number of rare translocation events, coupled with leakage, take places over a broad time range. When both membranes are saturated with peptides (exact time not determined), the cell wall breaks down, leading to outer membrane vesicle (OMV) formation ($\Delta t > 10$ s), detachment of outer and inner membranes and waving ($\Delta t > 25$ min). Simultaneously, AMPs accumulate in internal compartments and reach saturation levels within less than 1 hr.

imply that AMPs are able to pass through inner and outer membranes without any noticeable effect on membrane structure, such as the transient formation of pores, nor does it exclude their presence. It merely refers to a generic, unspecified uptake of the peptides beyond the resolution of the present experiments. Further, since we are not able to discriminate membrane peptide adsorption kinetics from the onset of leakage of cellular content, we cannot comment on a potential 'carpet mechanism,' proposed from analogous AMP partitioning studies in bacteria (*Roversi et al., 2014*; *Savini et al., 2020*). We note, however, that LF11-215 was found to partition into artificial lipid membranes, without noticeable effects on lipid flip-flop (*Marx et al., 2021a*). We detail this discussion further below.

The drop of $R$ is a natural consequence of the loss of cellular content, but occurs at somewhat later times. This delay is most likely due to the stored elastic energies of the peptidoglycan layer, which will initially resist rapid deformations (*Yang et al., 2018*). Note, however, that the peptidoglycan properties are likely affected by direct interactions with peptides (*Zhu et al., 2019*). Remarkably, cytoplasmic membrane leakage in the MIC range occurs ~1 min later for LF11-215 (*Figure 2—figure supplement 3*).

Pronounced differences between LF11-324 and LF11-215 were also observed from their efficacies as a function of cell concentration (*Figure 4*). At equal $n_{cell}$, growth-inhibition probability distributions are much narrower for LF11-324. Apparently, this increased 'cooperativity' correlates with the peptide's ability to permeabilize the cytoplasmic membrane faster. It is further illuminating to discuss the total amount of peptide penetrating into the intracellular space. Both peptides saturate the outer

LPS leaflet already at concentrations lower than 0.3× MIC (*Figure 5—figure supplement 1*), corresponding to about one AMP per 4–5 LPS molecules as upper-bound value (see $N_P^{max}/N_{LPS}^0$ in 'Materials and methods'). Thus, consistent with *Figure 2A and B*, peptides penetrate the outer membrane already when the effects on bacterial growth are still very small. Assuming that the inner leaflet of the outer membrane, as well as the two leaflets of the inner membrane, host to first-order similar amounts of LF11-324 and LF11-215, we calculate that ≥92% of all peptides are located in the intracellular space at the respective MICs. Using our USAXS/SAXS/DLS data for the average cytoplasmic and periplasmic volumes, we further arrive at huge intracellular concentrations of ~80 mM for LF11-324 and ~100 mM for LF11-215. The relative difference in cytosolic/periplasmic concentrations between the two peptides corresponds to about the relative difference in MICs (*Figure 1—figure supplement 1*). Our finding that only a minor fraction of the peptides is located within the membranes remains valid even at sub-MIC values, offering an attractive explanation for the similar and concentration-independent average cell-wall damage-induced LF11-324 and LF11-215. Specifically, we propose that this damage (LPS packing, OMV formation, membrane ruffling and detachment) is a consequence of the small fraction of peptides that saturate the membranes and, possibly, the peptidoglycan layer. Following the thermodynamic principles of the partitioning framework and $\zeta$-potential measurements, this small fraction is supposed to be rather constant above AMP concentrations of about 0.2× MIC. Along these lines, recent in-cell FRET experiments showed that only a small fraction of the peptides partitioned in a cell is actually interacting with the bacterial surface (*Kaji et al., 2021*).

About 10 years ago, *Wimley, 2010* already discussed the consequences of having $10^7$ to $10^9$ AMPs per cell, suggesting a 'reservoir' of non-membrane-bound peptides that would outnumber proteins, ATP, and other metabolites. Cytosolic targets were also confirmed by our TEM data (and those of others; *Hammer et al., 2010*; *Scheenstra et al., 2019*) showing a collapsed nucleoid region (*Figure 1*). For instance, this could be due to a competition mechanism with polyamines as putative stabilizers of the functional architecture of the DNA ring (*Hou et al., 2001*). In support of the hypothesis of cytosolic targets, recent solid-state $^{31}$P-NMR measurements of *E. coli* in the presence of AMPs revealed increased dynamics of DNA and RNA phosphate groups correlated to TEM observations of collapsed nucleoid volume (*Overall et al., 2019*). In addition, however, AMP interactions with other negatively charged metabolites and macromolecules in the cytosol, such as ribosomes and proteins, should be considered as potentially detrimental to bacteria (*Zhu et al., 2019*). The about 1.4 times lower intracellular concentration of LF11-324 then is a signature of a higher potency compared to LF11-215, possibly hampering a number of physiological functions (*Scocchi et al., 2016*), or inducing an apoptosis-like mechanism (*Dwyer et al., 2012*) due to interactions with any of the above-named cytosolic molecules. The exact killing mechanism of the here-studied AMPs, however, evades current experimental design.

It follows from the considerations above that bacteria have an increased probability to recover from the peptides' attack if the intracellular concentrations of LF11-324 and LF11-215 fall below those reported above (i.e., $\lesssim 80$ and $\lesssim 100$ mM, respectively). Thus, both the ability of LF11-324 to swiftly permeabilize the cytoplasmic membrane by translocating through the cell envelope and its higher propensity to interfere with physiological processes contribute to its higher 'cooperativity' in killing *E. coli* (*Figure 4*). Our experimental setup is not sensitive to directly observe either transient membrane pores or other membrane defects. Note, however, that previous studies in lipid-only mimics of bacterial membranes showed only weak membrane remodeling effects of lactoferricin derivatives compared to other peptides (*Zweytick et al., 2011*; *Marx et al., 2021b*). We propose that the here-studied LF11s follow the interfacial activity hypothesis, valid for both antimicrobial (*Wimley, 2010*) and cell-penetrating peptides (*Kauffman et al., 2015*). That is, they are able to locally and transiently induce disorder in the lipids' hydrocarbon chain regime, which sufficiently affects the permeability barrier of the bilayer against ions, small polar molecules, and, eventually, the peptides themselves. This might be also a consequence of the rather short amino acid sequence of the lactoferricin derivatives, implying a too small length (~1 nm) to span the whole membrane thickness (3–4 nm). In turn, the highly flexible secondary structure and an amphipathic moment being aligned along the peptide's backbone (*Zorko et al., 2009*; *Zweytick et al., 2011*; *Zweytick et al., 2014*) should allow LF11-324 and LF11-215 to translocate membranes at higher rates than observed for linear peptides (*Ulmschneider, 2017*; *Kabelka and Vácha, 2018*). Note that peptide translocation can also lead to transient membrane leakage events (*Ulmschneider, 2017*), even with negligible AMP-induced lipid

flip-flop (*Marx et al., 2021a*). The higher hydrophobicity of O-LF11-215 should increase the likelihood of remaining membrane bound, which might build up differential membrane curvature stress and lead to the observed formation of membrane tubules. We also note that the different leakage kinetics for the LF11 peptides suggest a coupling to translocation kinetics, which in turn depends on membrane partitioning of the AMPs. Indeed, recently reported data for cytoplasmic membrane mimics of cardiolipin, phosphatidylethanolamine, and phosphatidylglycerol show a somewhat faster membrane partitioning for LF11-324 than for LF11-215 (*Marx et al., 2021b*).

In conclusion, the superior time resolution and sensitivity to *ensemble* averaged structural changes from cellular size to molecular packing of synchrotron USAXS/SAXS allowed us to demonstrate, upon combination with advanced data modeling and complementary techniques for peptide partitioning, that LF11-based AMPs are able to reach the cytosolic membrane of bacteria on the seconds time scale and subsequently accumulate at high concentrations (80–100 mM) in all bacterial compartments, including the cytosol. Based on differences observed for LF11-324 and LF11-215, two factors emerge as key components for their antimicrobial efficacy: (i) a fast translocation through inner and outer membranes, and (ii) an efficient shutdown of here not further detailed vital chemical/biochemical processes, that is the lower the number of 'needed' AMPs the better (here achieved by LF11-324). The latter conclusion is corroborated by the most striking fact that the final cell-wall damage *does not depend* on peptide concentration. That is, AMPs inhibiting only 1% of bacteria induce on average—besides some single isolated variations—as much membrane ruffling, detachment of inner and outer membranes, leakage, etc., as AMPs at full levels of growth inhibition. Hence, the collateral damage of the cell-wall structure, including leakage of cellular content, is a 'by-product' of AMP activity. Instead, we suggest that the primary cause of growth inhibition is driven by interactions of the AMPs with yet-to-be-determined cytosolic/periplasmic molecules. Most likely candidates are the polyanionic DNA, RNA, ribosomes, and proteins (*Zhu et al., 2019*), or charged metabolites. We strongly emphasize that the ability of the here-studied peptides to impair the barrier function of bacterial membranes remains an unquestioned highly important property. However, at the same time our study shows that this effect alone is not sufficient to explain all of our data.

It is currently not clear whether the present findings can also be extended to other AMPs. Yet, the importance of the cell-penetrating ability was similarly discussed for, for example, peptides Sub3 and LL-37 (*Torcato et al., 2013*; *Zhu et al., 2019*). Further, a number of AMPs have been reported to show partitioning behavior in bacteria and accumulation in the intracellular volume analogously to the here-studied lactoferricin derivatives (see, e.g., *Sochacki et al., 2011*; *Zhu et al., 2019*; *Savini et al., 2020*; *Loffredo et al., 2021*; *Kaji et al., 2021*, and references therein). *Le et al., 2017* gave a more complete account of potential intracellular targets of AMPs—either of fundamental or secondary importance for bactericidal activity. We thus propose that the combination of membrane translocation speed and efficient impairment of physiological processes are generic factors that should be considered in designing future AMPs to combat infectious diseases. This also implies a widening of the pure focus on membrane activity of AMPs currently applied in many studies.

## Materials and methods
### Samples
*E. coli* ATCC 25922 were provided by the American Type Culture Collection (Manassas, VA). Freeze-dried peptide powders of LF11-215 (H-FWRIRIRR-NH$_2$), LF11-324 (H-PFFWRIRIRR-NH$_2$), and O-LF11-215 (octanoyl-FWRIRIRR-NH$_2$), purity >95%, were purchased from the Polypeptide Laboratories (San Diego, CA). Lysogeny broth (LB)-agar and LB medium were obtained from Carl Roth (Karlsruhe, Germany). All the other chemicals were purchased from Sigma-Aldrich (Vienna, Austria).

### Bacterial cultures
Bacterial colonies of *E. coli* ATCC 25922 were grown in LB-agar plates at 37°C. Overnight cultures (ONCs) were prepared by inoculating a single colony in 3 ml LB-medium in sterile polypropylene conical tubes (15 ml), allowing for growth under aerobic conditions for 12–16 hr in a shaking incubator at 37°C. Main cultures (MCs) were then prepared by diluting ONCs in 10 ml LB-medium in 50 ml sterile polypropylene conical tubes. Bacteria in the MCs grew under the same conditions applied to ONCs up to the middle of the exponential growth phase. Cells were then immediately washed twice and

resuspended in nutrient-free and isotonic phosphate-buffered saline (PBS) solution (phosphate buffer 20 mM, NaCl 130 mM) at pH 7.4.

## AMP samples

LF11-324 and O-LF11-215 peptides displayed a weak solubility in PBS. AMP stocks (including LF11-215) were then prepared by adding acetic acid and DMSO, up to 0.3 and 3% vol/vol, respectively. Peptide stock solutions were diluted for measurements yielding a final concentration of 0.01% acetic acid and 0.1% vol/vol dimethyl sulfoxide (DMSO) (final pH 7.2). Hence, possible effects of DMSO on the cell envelope, as observed for model membrane structures (*Gironi et al., 2020*), can be neglected. Control USAXS/SAXS experiments adding a similar amount DMSO and acetic acid to *E. coli* also showed no discernible effect of the organic solvent (data not shown). Similarly, control experiments were performed to exclude effects on bacterial growth. Stock concentrations were determined by measuring the absorption band of the Trp residues with the spectrophotometer NanoDrop ND-1000 (Thermo Fisher Scientific, Waltham, MA). Peptide stock solutions were stored in silanized glass tubes until use.

## Antimicrobial activity and partitioning modeling

The antimicrobial activity of the AMPs on *E. coli* was tested in the bacterial concentration range of $5 \times 10^5$ to $10^9$ CFU/ml using a modified susceptibility microdilution assay (*Jorgensen and Ferraro, 2009*). Cell suspensions were incubated with AMPs in buffer at a given peptide and cell concentration for 2 hr at 37°C (control samples were incubated in buffer only). Cell growth was monitored upon addition of double concentrated LB-medium for about 20 hr using a Bioscreen C MBR (Oy Growth Curves Ab, Helsinki, Finland).

### Partitioning modeling

The analysis of the inhibited fraction of cells, $\phi_{IG}$, as a function of the total concentration of peptides, $[P]$, enabled the extraction of $IC_x$ values as a function of $n_{cell}$, as detailed in Appendix 3. Following previously reported methods to determine peptide partition in liposomes (*White et al., 1998*) or bacterial systems (*Savini et al., 2017*), we derived the number of cell partitioned AMPs for a given $IC_x$ value using

$$[P](n_{cell}) = \underbrace{\frac{N_B[W]}{K^{eff}}}_{[P]_W} + \underbrace{\frac{N_B}{N_A}n_{cell}}_{[P]_B} = \frac{N_B}{N_A}\left(\frac{N_A[W]}{K^{eff}} + n_{cell}\right),$$

(1)

where $[P]_W$ and $[P]_B$ are the concentrations of AMPs dispersed in the aqueous phase and segregated into the cells, respectively; $N_A$ is the Avogadro's constant; $[W]$ is the concentration of water molecules in bulk (55.3 M at 37°C); $K^{eff}$ is the effective mole-fraction partitioning coefficient; and $N_B$ is the *average* number of peptide monomers that are partitioned within a single cell. Details of this analysis are reported in *Marx et al., 2021b*.

A similar approach was exploited in the case of O-LF11-215 peptide clusters in solution. In this case, the total peptide concentration is given by $[P] = [P]_W + [P]_B + N[A]$, where $[A]$ is the molar concentration of aggregates, each of them consisting of an average number of peptides $N$. We also define the aggregate fraction $f_A = N[A]/[P]$ and assume the equilibrium state $N[P]_W \rightleftharpoons [A]$. The definition of the molar partitioning coefficient $K_x \propto K^{eff}$ (*Marx et al., 2021b*) refers to the balance of concentration of free peptides in bulk and partitioned peptide into the cells. Hence, its bare definition is unaffected by the presence of clusters. Finally, it is trivial to show that in this case *Equation 1* becomes

$$[P](n_{cell}) = \frac{N_B}{N_A(1-f_A)}\left(\frac{N_A[W]}{K^{eff}} + n_{cell}\right) = \frac{N_B^{eff}}{N_A}\left(\frac{N_A[W]}{K^{eff}} + n_{cell}\right),$$

(2)

where the fitting parameter $N_B^{eff}$ is an upper boundary value for the actual number of partitioned peptides per cell $N_B = N_B^{eff}(1 - f_A)$.

## ζ-potential, cell size, and outer leaflet distribution of peptides

$\zeta$-potential and dynamic light scattering (DLS) measurements were carried out with the Zetasizer Nano ZSP (Malvern Panalytical, Malvern, UK). *E. coli* suspensions were incubated with different concentrations of AMPs in buffer for 1 hr at 37°C prior to each measurements. Control samples (no AMPs) were suspended and incubated in buffer. Measurements were also repeated just after mixing with peptides (about 5 min waiting time for sample loading, data not shown), exhibiting negligible variation from samples incubated for 1 hr. A concentration of $10^7$ CFU/ml provides the optimal compromise between high signal-to-noise ratio and low multiple-scattering bias. The AMP concentrations were centered on the MIC values previously determined with the susceptibility microdilution assays, spanning from about 0.2× to 2.5× MIC. For $\zeta$-potential measurements, the voltage for the electrodes was set to 4 V, such that currents did not exceed 1 mA, because of the high conductivity of the PBS buffer. Further, measurements were paused between repetitions for 180 s. This prevented heat productions, leading to sample denaturation and accumulation on the electrodes. The experiments were repeated three times, and, due to the low sensitivity of such a setup, each of them consisted of a minimum of six measurements (see also *Marx et al., 2021b*). For each system, $\zeta$-potential values and associated errors were given by the medians and the median absolute deviations, respectively, averaging over at least 18 repetitions. The same number of scans was also used to obtain mean and standard deviation values for the hydrodynamic diameter, $d_H$, of the cells.

From the measured $\zeta$-potential values, we calculated the maximum number of peptides that partition into the outer LPS leaflet, $N_P^{max}$, as reported recently (*Marx et al., 2021b*)

$$\frac{N_P^{max}}{N_{LPS}^0} \approx \left( \frac{z_{LPS}}{z_P} \right) \left( \frac{\zeta}{\zeta_0} \frac{S}{S_0} - 1 \right), \tag{3}$$

where $z_{LPS} = -6$ (*Wiese et al., 1998*) and $z_P = +5$ (*Zweytick et al., 2011*) are the nominal charges of LPS and LF11-215 or LF11-324 AMPs, respectively; $\zeta$ and $S$ are the $\zeta$-potential and total surface values of the system upon addition of peptides; and $\zeta_0$ and $S_0$ are the respective reference values (no AMPs). $N_{LPS}^0 \approx 0.9 S_0 / A_{LPS}$ is the estimated number of LPS molecules, where $A_{LPS} \simeq 1.6\,\text{nm}^2$ (*Kim et al., 2016*; *Micciulla et al., 2019*) is the lateral area per LPS molecule. The prefactor originates from considering a maximum surface coverage of 90% by LPS molecules (*Seltmann and Holst, 2002*). $S_0$ was derived from DLS measurements, approximating the bacterial shape by a cylinder $d_H/2 \approx \sqrt{(\text{radius})^2/2 + (\text{length})^2/12}$, considering that the hydrodynamic radius is approximately equivalent to the radius of gyration for micron-sized objects. Then fixing the radius at ~400 nm (*Semeraro et al., 2021b*) and retrieving the length from the above relation for $d_H$ one obtains $S_0 \approx 5 \times 10^6\,\text{nm}^2$.

## Fluorescence spectroscopy

Fluorescence spectroscopy experiments were done with the Cary Eclipse Fluorescence Spectrophotometer (Varian/Agilent Technologies, Palo Alto, CA). The excitation wavelength was set to $\lambda = 280$ nm (which corresponds to the maximum intensity of the absorption/excitation band of Trp), and emission spectra were acquired in the $\lambda$-range between 290 and 500 nm, with the Trp emission band peak being expected to lie around 330–350 nm. Samples were loaded in quartz cuvettes of 1 cm path length. The background was subtracted from every Trp spectrum prior to further analysis.

### Peptide solubility

Trp emission allowed determining whether LF11 peptides form aggregates or not in the MIC range. Spectra of LF11-only samples at $[P] = 100$ μg/ml were fitted with the log-normal-like function (*Burstein and Emelyanenko, 1996*; *Ladokhin et al., 2000*):

$$I(I_0, \lambda, \Gamma) = \begin{cases} I_0 \exp\left[ -\frac{\ln 2}{\ln^2 \alpha} \ln^2\left( 1 + \frac{(\lambda - \lambda_{max})}{y\Gamma} \right) \right], & \lambda > (\lambda_{max} - y\Gamma) \\ 0, & \lambda \leq (\lambda_{max} - y\Gamma) \end{cases} \tag{4}$$

where $\lambda_{max}$ and $I_0$ are, respectively, wavelength and intensity of the emission peak; $\Gamma$ is the FWHM of the band; $\alpha$ is a skewness parameter (fixed at an optimum value of 1.36 after testing; and $y = \alpha/(\alpha^2 - 1)$).

Both LF11-215 and LF11-324 (see Appendix 1) showed a peak at about $\lambda_{max} \simeq 353$ nm and $\Gamma \sim 63$ nm. This is consistent with a location of the Trp residues in polar chemical environments having full

mobility and thus suggests that these AMPs are monomeric (**Burstein and Emelyanenko, 1996**). In contrast, the acylated O-LF11-215 showed a significant blue shift related to a location of Trp within apolar surroundings (**Ladokhin et al., 2000**), indicating the formation of peptide aggregates.

## Peptide partitioning

Analogously to the partitioning analysis performed for lipid-only membranes (**Marx et al., 2021b**), the Trp emission of bacteria/AMPs mixtures were treated as a two-component signal, one coming from the peptides in the aqueous phase, and the second one from AMPs interacting with the cells.

Bacterial suspensions were incubated with different concentrations of AMPs in buffer for 1 hr at 37°C (incubator Thermomixer C, Eppendorf, Germany). Reference samples (no AMPs) were suspended and also incubated in PBS. Experiments were performed at cell concentrations of $5 \times 10^7$, $10^8$, and $5 \times 10^8$ CFU/ml, and AMPs amounts equal to 0.5×, 1×, and 2× MIC (each experiment was repeated three times). Fluorescence intensities were background-subtracted using the bacteria-only reference spectra at the corresponding concentrations. This enabled us to subtract the average signal from the aromatic residues in the cells, and the scattering arising from the high concentration of the cell suspensions. Emission spectra of bare bacterial systems surprisingly showed a single and clean Trp-emission band, peaking at $\lambda_{max} = 334 - 336$ nm ($\Gamma \sim 65$ nm). The peak intensity increased linearly from 4.9 (arb. units) at $10^7$ cells/ml to 122 (arb. units) at $5 \times 10^8$ cells/ml; then turbidity effects had to be accounted for. LF11-215 and LF11-324 showed peaks of comparable intensities even in the high cell-concentration regime, enabling us to retrieve background-subtracted spectra of optimal signal-to-noise ratio. Spectra were analyzed with a linear combination of two independent bands (see **Equation 4**) $I^W$ and $I^B$, referring to AMPs in bulk (W) and partitioned into the lipid bilayer (B). $\lambda^W$ and $\Gamma^W$ were fixed to the reference values obtained by analyzing spectra from pure AMPs, and $I_0^W$, $I_0^B$, $\lambda^B$, and $\Gamma^B$ were freely adjusted. LF11-only solutions were measured to calibrate their intensity dependence in buffer. Then, the retrieved $I_0^W$ values were converted to the concentration of dissociated peptides $[P]_W$. This allowed us to obtain the so-called peptide bound fraction as $f_B = 1 - [P]_W/[P]$ (see Appendix 4). The aggregation of O-LF11-215 led to low $\lambda_{max}^W$ values (see Appendix 1) and precluded a similar analysis for these peptides.

## Transmission electron microscopy

*E. coli* suspensions at $5 \times 10^8$ CFU/ml were mixed with peptides at the corresponding MICs and 0.5× MICs in PBS, and incubated for 1 hr at 37°C (incubator Thermomixer C, Eppendorf, Germany). Control samples (no AMPs) were suspended and incubated in PBS. Cell suspensions were centrifuged at 1300 × $g$ for 4 min in 1.5 ml Eppendorf tubes and resuspended (fixed) in 3% vol/vol glutaraldehyde and 0.1 M cacodylate buffer to bring the rapid cessation of biological activity and to preserve the structure of the cell. After fixation, the samples were washed and postfixed in 1% vol/vol $OsO_4$ in 0.1 M cacodylate buffer. Dehydration was carried out in an ascending ethanol series followed by two steps with propylene oxide (**Hayat, 1989**) and embedded in Agar Low Viscosity Resin (Agar Scientific, Stansted, UK). Ultrathin sections (70–80 nm) were prepared on an Ultramicrotome UC6 (Leica Microsystems, Vienna, Austria) equipped with a 35° Ultra Diamondknife (Diatome, Nidau, Switzerland). The grids were poststained with Uranyless (Science Services, Munich, Germany) and lead citrate according to Reynolds (**Hayat, 1989**). TEM images were acquired with Tecnai T12 at 120 kV (TFS, Warmond, Netherlands).

## Small-angle scattering

### (Ultra-) small-angle X-ray scattering

USAXS/SAXS measurements were performed on the TRUSAXS beamline (ID02) at the European Synchrotron Research Facility (ESRF), Grenoble, France. The instrument uses a monochromatic beam ($\lambda$ = 0.0995 nm) that is collimated in a pinhole configuration. Measurements were performed with sample-to-detector distances of 30.8 and 3.0 m, covering a $q$-range of 0.001–2.5 nm$^{-1}$ (**Narayanan et al., 2018**). Two-dimensional scattering patterns were acquired on a Rayonix MX170 detector, normalized to absolute scale, and azimuthally averaged to obtain the corresponding one-dimensional USAXS/SAXS profiles. The normalized cumulative background from the buffer, sample cell, and instrument were subtracted to obtain the final $I(q)$. Bacterial samples ($n_{cell} \sim 10^9$ CFU/ml) were incubated with peptides in buffer for 1 hr at 37°C and directly measured in a quartz capillaries of 2 mm diameter

(37°C), mounted on a flow-through setup in order to maximize the precision of the background subtraction. Time-resolved experiments were instead performed with a stopped-flow rapid mixing device (SFM-3/4 Biologic, Seyssinet-Pariset, France), with 50 ms mixing of bacterial and peptides stock suspensions (37°C), and enabling data acquisition after a kinetic time of about 2.5 ms (*Narayanan et al., 2014*). A total of 50 frames was recorded for each experiment with an exposure time of 0.05 s and a logarithmic time-spacing ranging from 17.5 ms to about 10 min. Radiation damage tests were performed on reference systems prior to setting this X-ray exposure times. The scattering intensities were further corrected for sedimentation and background scattering from the stopped-flow cell.

## Contrast-variation small-angle neutron scattering

SANS experiments were performed on the D11 instrument at the Institut Laue-Langevin (ILL), Grenoble, France, with a multiwire $^3$He detector of 256 × 256 pixels (3.75 × 3.75 mm$^2$). Four different setups (sample-to-detector distances of 2, 8, 20.5, and 39 m with corresponding collimations of 5.5, 8, 20.5, and 40.5 m), at a wavelength $\lambda$ = 0.56 nm ($\Delta\lambda/\lambda$ = 9%), covered a $q$-range of 0.014–3 nm$^{-1}$. Two distinct *E. coli* suspensions were incubated with peptides LF11-215 and LF11-324 in buffer for 1 hr at 37°C. The bacterial concentration during the incubation was 10$^9$ CFU/ml, and both peptide concentrations were in the range of the measured MICs. Samples were then washed and resuspended in five different PBS solutions, containing 10, 30, 40, 50, or 90 wt% D$_2$O. Samples ($n_{cell} \sim 10^{10}$ CFU/ml) were contained in quartz Hellma 120-QS banjo-shaped cuvettes of 2 mm pathway and measured at 37°C. Cuvettes were mounted on a rotating sample holder, which prevented the bacteria from sedimenting. Data were reduced with the Lamp program from ILL, performing flat field, solid angle, dead time, and transmission correction. Further data were normalized by the incident neutron flux (via a monitor) and corrected by the contribution from an empty cell. Experimental setup information and data are available at https://doi.ill.fr/10.5291/ILL-DATA.8-03-910 .

Note that the present experimental time (~2 hr) is much shorter than the onset of cell lysis (*Zweytick et al., 2011*). As control, SANS measurements were repeated at extended times after mixing with the AMP in order to test for sample stability (data not shown), in terms of shape, cell-wall structure, and densities. The scattering intensities after 2, 4, 6, and 8 hr were comparable (with the exception of a weak decrease of $\rho_{CP}$ between 2 and 4 hr). Further, a comparison with TEM and SAXS data suggests that the peptide-induced cell damage does not progress any further after 1 hr.

## Data analysis: Peptide clusters

Reference O-LF11-215 samples were measured in the MIC range to investigate the microstructure of the peptide clusters. The SAXS pattern of O-LF11-215 was fitted with the equation

$$I_{\text{clu}}(q) = I^0_{\text{AMP}} e^{-(qR_g)^2/3} \left(1 + I_P q^{-f}\right), \tag{5}$$

where the term in brackets is related to the structure of the aggregates, and the exponential Guinier function accounts for the form factor of their subunits of radius of gyration $R_g$ (*Zemb and Lindner, 2002*), and $I^0_{\text{AMP}}$ is forward scattering intensity. The function $I_P q^{-f}$ is the Porod law that describes the high-$q$ asymptotic trend of scattering signal from the aggregates (*Glatter et al., 1982*), where $I_P$ is a scaling factor that depends on the surface properties of the aggregates, and $f$ is related to their fractal dimension (*Sorensen, 2001*; see Appendix 1).

## Data analysis: Bacterial modeling

X-ray and neutron scattering data were jointly analyzed with a recently reported analytical scattering model (*Semeraro et al., 2021b*). USAXS/SAXS patterns of end states displayed an excess scattering contribution between $q \sim$ 0.1–0.2 nm$^{-1}$ in the case of LF11-215 and LF11-324, not visible in the corresponding SANS patterns. Note that SANS experiments were conducted on samples that were washed and resuspended in different D$_2$O-containing buffer, while SAXS data were acquired immediately after 1 hr incubation with peptides. Together with the observation of OMVs by TEM (*Figure 1*), this suggests that the additional scattering contribution in SAXS data could be due to freely diffusing OMVs in the suspension medium.

All scattering data were fitted with the analytical functions

**Table 2.** List of fixed parameters for the combined analysis of USAXS/SAXS and contrast variation SANS data of *E. coli*.

| Description | Fixed parameters | Values |
|---|---|---|
| Center-to-center distance between the head-group layers in the CM | $D_{CM}$(nm) | 3.73 |
| Center-to-center distance between the head-group layers in the OM | $D_{OM}$(nm) | 3.33 |
| Width of the head-group layers for both CM and OM | $W_{ME}$(nm) | 0.75 |
| Center-to-center distance between the PG layer and the OM | $\Delta_{PG}$(nm) | 16.7 |
| Width of the PG layer | $W_{PG}$(nm) | 6.0 |
| Average SLD of the tail group layer in the CM | $\rho_{TI}$(nm$^{-2}$) $\times 10^{-4}$ | 8.31*/0.022† |
| Average SLD of the tail group layer in the OM | $\rho_{TO}$(nm$^{-2}$) $\times 10^{-4}$ | 8.86*/0.012† |
| Ratio between major and minor radii | $\epsilon$ | 2.0 |
| Effective radius of gyration of each OS core | $R_{g,OS}$(nm) | 0.45 |

USAXS/SAXS: (ultra) small-angle X-ray scattering; SANS: small-angle neutron scattering; SLD: scattering length density.
CM: Cytoplasmic membrane; OM: Outer membrane; PG: peptidoglycan; OS: Oligosaccharides.
*X-ray SLDs.
†Neutron SLDs.

$$I(q) = \begin{cases} I_{cell}(q) & \text{[Neutron data]} \\ I_{cell}(q) + n_{cell}I^0_{OMV}\left[3j_1(qR_{OMV})\right]^2 & \text{[X-ray data]} \\ I_{cell}(q) + I_{clu}(q) & \text{[O-LF11-215 data]} \end{cases} , \qquad (6)$$

where $I_{cell}(q)$ is the scattering form factor for *E. coli*, as reported in **Semeraro et al., 2021b**, and $3j_1(qR_{OMV})$ is the form factor of a sphere of radius $R_{OMV}$, with $j_1$ being the normalized spherical Bessel function of order 1. The prefactor $I^0_{OMV} = N_{OMV}V^2_{OMV}\Delta\rho^2_{OMV}$ is the OMV forward scattering, where $N_{OMV}$, $V_{OMV}$, and $\Delta\rho_{OMV}$ are, respectively, their number, volume, and SLD difference to the buffer. The choice of a simple spherical form factor was driven by its simplicity for checking whether bacteria and OMVs have to be considered as noninteracting scatterers or not. Tests using an interaction cross-term approximating budding OMVs by spheres decorating a larger surface (**Larson-Smith et al., 2010**) did not result in significant contributions, confirming the dominance of freely diffusing OMVs. Note also that the $I^0_{OMV}$ values were independent of the shape of the normalized form factor as they include our estimation of $V_{OMV}$ and $\Delta\rho_{OMV}$ (Appendix 2). Finally, in the case of SANS data, instrumental smearing was taken into account. Data were fitted with a convolution of $I(q)$ and a Gaussian function with *q*-dependent width values, as provided by the reduction tools at D11. In USAXS/SAXS data, the smearing effect was negligible.

**Table 3.** List of fixed and D$_2$O-dependent parameters for the combined analysis of USAXS/SAXS and contrast variation SANS data of *E. coli*.

The average SLD of both CM and OM head-group layers, $\rho_{ME}$, the SLD of the buffer solution, $\rho_{BF}$, and the product of the each OS core volume and its contrast relative to the buffer, $\beta_{OS} = V_{OS}\Delta\rho_{OS}$.

| Fixed parameters | Values | | | | | |
|---|---|---|---|---|---|---|
| | | Neutrons (wt% D$_2$O) | | | | |
| | X-rays | 10 | 30 | 40 | 50 | 90 |
| $\rho_{ME}$(nm$^{-2}$) $\times 10^{-4}$ | 12.9 | 1.56 | 2.20 | 2.52 | 2.84 | 4.11 |
| $\rho_{BF}$(nm$^{-2}$) $\times 10^{-4}$ | 9.476 | 0.135 | 1.54 | 2.20 | 2.81 | 5.54 |
| $\beta_{OS}$(nm) $\times 10^{-4}$ | 10.7 | 3.83 | 2.32 | 1.68 | 0.69 | –2.44 |

USAXS/SAXS: (ultra) small-angle X-ray scattering; SANS: small-angle neutron scattering; SLD: scattering length density.

After thorough testing, the analysis of SAS data (end states and kinetics) was conducted using only seven adjustable parameters describing $I_{cell}(q)$. These were the number of LPS molecules, $N_{OS}$; the cytoplasm radius, $R$; the SLDs of the cytoplasmic, $\rho_{CP}$, and periplamic space, $\rho_{PP}$; the periplasmic average thickness, $\Delta_{OM}$, and its deviation, $\sigma_{OM}$; and the SLD of the peptidoglycan layer, $\rho_{PG}$. Additionally, $I^0_{OMV}$ and $R_{OMV}$ were fitted for scattering intensities in the presence of LF11-215 and LF11-324, while $I^0_{AMP}$ was used and adjusted in the case of O-LF11-215. Other parameters of $I_{clu}(q)$ were fixed according to the O-LF11-215 alone systems (see *Table 1*). This allowed us to fully describe the scattering-pattern variations upon peptide activity. Other parameters, including those accounting for the structure of inner and outer membranes, were fixed to the references values (see complete description in *Semeraro et al., 2021b*; all fixed parameters are listed in *Tables 2 and 3*).

The scattering intensities of O-LF11-215-aggregates were comparable to that of bacteria in the high *q*-range (*Figure 2—figure supplement 1D*). While this affected the quality of the ultrastructural parameters, it enabled at the same time the investigation of the kinetics of the AMP uptake. Indeed, by assuming that O-LF11-215 is primarily forming aggregates in solution, $I^0_{AMP}$ at $\Delta t = 17.5$ ms can be converted to the total known peptide concentration $[P]$. Hence, the further assumption that peptides leaving the clusters are directly partitioning into the cell allows to convert the difference $\left[ I^0_{AMP}(\Delta t) - I^0_{AMP}(0) \right]$ to $[P]_B(\Delta t)$. It follows that

$$N^{kin}_B(\Delta t) = \frac{N_A [P]_B(\Delta t)}{n_{cell}}, \tag{7}$$

where $N^{kin}_B$ is the number of O-LF11-215 partitioned within the volume of a single cell that can be obtained time-resolved USAXS/SASX data.

Finally, time-resolved USAXS/SAXS data were fitted using the parameters of the initial (see *Semeraro et al., 2021b*) and end states as boundaries and guide to refine the $\chi^2$ minimization (see *Figure 2—figure supplement 4*). This was accomplished by means of a genetic selection algorithm exploiting >300 repetitions of converging fittings (see details in *Semeraro et al., 2021b*). Fit results were filtered using an ad hoc threshold of $\chi^2 < 1.15\chi^2_{min}$, which enabled the exclusion of local minima. Filtered results were used to build up parameter distributions that in turn provided mean values, associated errors (standard deviation), and correlation coefficients. Exemplary distributions are shown in *Figure 2—figure supplement 5* and *Figure 2—figure supplement 6*. Variations in $\Delta_{OM}$ and $\sigma_{OM}$ at $\Delta t = 17.5$ ms are due to lower signal-to-noise ratio available in time-resolved measurements.

## Acknowledgements

ESRF – The European Synchrotron and the Institut Laue–Langevin (ILL) are acknowledged for provision of SAXS (proposals LS-2513 and LS-2869) and SANS (exp. 8-03-910) beamtimes. The authors are grateful to T Narayanan for his invaluable support, and thank JM Devos, D Marquardt, and M Pachler for their support during the proof-of-concept experiments (LS-2513), and the biological support laboratory at EMBL Grenoble for providing access to the laboratory equipment for bacterial sample preparation. The authors also acknowledge N Malanovic for sharing her expertise about bacterial cultures and S Keller for the fruitful discussions. Finally, the authors thank the whole staff of ID02 and the D11 for support and availability.

## Additional information

### Funding

| Funder | Grant reference number | Author |
|---|---|---|
| Austrian Science Fund | P 30921 | Karl Lohner |

The funders had no role in study design, data collection and interpretation, or the decision to submit the work for publication.

## Author contributions

Enrico F Semeraro, Conceptualization, Data curation, Formal analysis, Investigation, Methodology, Project administration, Software, Validation, Visualization, Writing – original draft, Writing – review and editing; Lisa Marx, Formal analysis, Investigation, Validation; Johannes Mandl, Data curation, Formal analysis, Investigation, Validation; Ilse Letofsky-Papst, Claudia Mayrhofer, Data curation, Investigation, Resources, Validation, Writing – review and editing; Moritz PK Frewein, Investigation; Haden L Scott, Investigation, Writing – review and editing; Sylvain Prévost, Data curation, Investigation, Resources; Helmut Bergler, Conceptualization, Funding acquisition, Investigation, Resources; Karl Lohner, Conceptualization, Funding acquisition, Project administration, Resources, Supervision, Writing – review and editing; Georg Pabst, Conceptualization, Funding acquisition, Investigation, Project administration, Resources, Supervision, Validation, Writing – original draft, Writing – review and editing

## Author ORCIDs

Enrico F Semeraro (iD) http://orcid.org/0000-0002-6096-1108
Moritz PK Frewein (iD) http://orcid.org/0000-0002-0329-5305
Sylvain Prévost (iD) http://orcid.org/0000-0002-6008-1987
Helmut Bergler (iD) http://orcid.org/0000-0002-7724-309X
Georg Pabst (iD) http://orcid.org/0000-0003-1967-1536

## Decision letter and Author response

Decision letter https://doi.org/10.7554/eLife.72850.sa1
Author response https://doi.org/10.7554/eLife.72850.sa2

## Additional files

### Supplementary files

• Transparent reporting form

### Data availability

The current manuscript is a biophysical study, reporting data analysis of scattering data and peptide partitioning assays in vitro. All relevant data are included and plotted in the manuscript. In addition SANS raw data are accessible (http://dx.doi.org/10.5291/ILL-DATA.8-03-910). The modelling code for data analysis consists of a standard chi-squared minimization algorithm. The implemented analytical functions are described in the Methods and Materials section of the manuscript.

The following dataset was generated:

| Author(s) | Year | Dataset title | Dataset URL | Database and Identifier |
| --- | --- | --- | --- | --- |
| Pachler M, Frewein MPK, Lohner K, Marx L, Pabst G, Prevost S, Haden S, Semeraro EF | 2022 | Antimicrobial peptide induced phase separation in *E. coli* membrane mimetic systems | http://dx.doi.org/10.5291/ILL-DATA.8-03-910 | Institut Laue-Langevin, 10.5291/ILL-DATA.8-03-910 |

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

## Appendix 1

### Clusters of acylated peptide O-LF11-215

Peptide clusters formed by O-LF11-215 were investigated by Trp fluorescence and USAXS/SAXS. Trp spectra displayed a $\lambda_{max} \simeq 336$ nm and an FWHM of about 67 nm, which can be related to a heterogeneous distribution of Trp with apolar surroundings (*Ladokhin et al., 2000*). In addition, O-LF11-215 exhibited Trp phosphorescence emission at about 450 nm, which usually is not measurable due to its dynamic quenching by oxygen and impurities in aqueous suspensions (*Cioni and Strambini, 2002*). Its presence suggests that a significant portion of Trp residues are buried in hydrophobic cores, with no access to the solvent and with a local high viscosity.

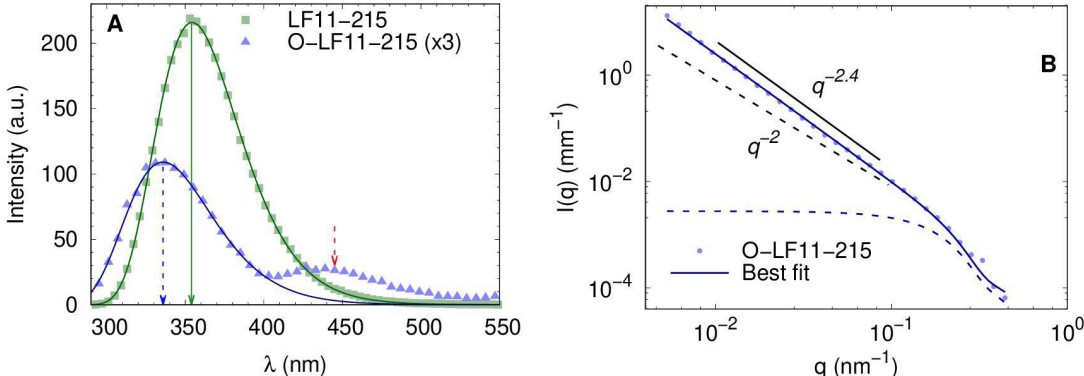

**Appendix 1—figure 1.** Trp-fluorescence and SAXS analyis of O-LF11-215 clustering. (**A**) Trp fluorescence data of LF11-215 (green squares) and O-LF11-215 (blue triangles) at 100 µg/ml (LF11-324 are not shown to avoid redundancy). Data were fitted with *Equation 4*. Arrows mark the maxima positions of the fluorescence and phosphorescence bands. (**B**) Small-angle X-ray scattering (SAXS) data of O-LF11-215 at 400 µg/ml. The fit was performed with *Equation 5*. Additional lines highlight the obtained intensity decay of slope –2.4 (solid line), typical of mass fractals, as opposed to –2.0 (dashed line), which would correspond to either Gaussian chains or planar structures. Note that the slope is conserved in a *q*-range larger than one order of magnitude.

USAXS/SAXS data in the low $q$-range ($q_{min} \sim 0.005$ nm$^{-1}$) exhibited a featureless decay of intensity with a slope of $f = 2.4$. This slope value is typical for mass fractals, that is, highly branched objects with high surface-to-volume ratio, while $q_{min}$ suggests a minimum aggregate size of $\sim 2\pi/q_{min} > 1\mu$. Furthermore, a Guinier term is needed to fit the shoulder at about $q = 0.2$ nm$^{-1}$ corresponding to an average radius of gyration $R_g \simeq 10$ nm. Note that this feature also does not vanish for different choices of scaling constants for background subtraction. Interestingly, this value is way too high to describe O-LF11-215 monomers, whose expected radius of gyration would be $lt_1$ nm. Possibly, peptide monomers create smaller aggregates of mean size $R_g \simeq 10$ nm, which in turn form a heterogeneous and branched supramolecular structure with the characteristics of a mass fractal.

## Appendix 2

### Estimating the scattering contribution of OMVs

The prefactor of the scattering contribution from extracellular, independent objects used in *Equation 6* is $I^0_{OMV} = N_{OMV} V^2_{OMV} \Delta\rho^2_{OMV}$, where $N_{OMV}$ is the number of OMVs, $V_{OMV}$ is the volume of an OMV, and $\Delta\rho_{OMV}$ is average SLD difference to the buffer. This definition of forward scattering (per single cell) is valid for every noninteracting object. Hence, to validate the assumption that this scattering contribution is related to OMVs, it is interesting to calculate possible $N_{OMV}$, $V_{OMV}$, and $\Delta\rho_{OMV}$ values. Note that even if modeling OMVs as homogeneous spheres appears as a crude first-order approximation, $R_{OMV}$ can be associated to its radius (within ~10% confidence). Assuming the same lipid asymmetry as in the outer membrane, the inner leaflet of OMVs can be mimicked by a 3:1 mole mixture of palmitoyl-oleoyl-phosphatidylethanolamine (POPE) and palmitoyl-oleoyl-phosphatidylglygerol (POPG), respectively (*De Siervo, 1969*; *Lohner et al., 2008*; *Leber et al., 2018*). The SLD membrane profiles of these lipids have been thoroughly investigated (*Kučerka et al., 2012*, *Kučerka et al., 2015*). The outer leaflet might instead be dominated by LPS, whose lipid A possesses about six short C14:0 chains (*Kim et al., 2016*), and the polar region can be approximated as two PG units, in terms of molecular volume and SLD. In addition, LPS inner and outer core volumes and SLDs can be calculated from *Heinrichs et al., 1998* and *Müller-Loennies et al., 2003*, neglecting O-antigen chains for simplicity. Gathering all this information, similarly to the membrane structure estimation in *Semeraro et al., 2021b*, the vesicles would have a membrane thickness of 4.1 nm and an average SLD of $1.1 \times 10^{-3}$ nm$^{-2}$ (volume-weighted averages). The lumen of OMVs can be quite diversely composed (*Beveridge, 1999*). We tentatively assigned the SLD of the periplasmic space of the end-state system, that is, $9.68 \times 10^{-4}$ nm$^{-2}$.

Together with a buffer SLD of $9.47 \times 10^{-4}$ nm$^{-2}$, a measured $I^0_{OMV} = 1500 \pm 200$ nm and a $R_{OMV} = 15.4 \pm 0.6$ nm, for instance, then leads to the estimate $N_{OMV} = 1200 \pm 400$ and a total lipid surface (inner and outer leaflets of all OMVs) of $(4 \pm 2) \times 10^6$ nm$^2$.

## Appendix 3

### Statistics of bacterial inhibition

Assuming that the AMP-induced delayed bacterial growth is entirely due to a lower number density of survived cells (**Marx et al., 2021b**), the inhibited fraction of cells, $\phi_{IG}$, as a function of peptide and cell concentrations was fitted with a heuristic approach. Specifically, we used the sigmoidal Gompertz function

$$F([P]; b, c) = \exp[-b \exp(-c[P])], \tag{8}$$

where $[P]$ is the total peptide molar concentration, and $b$ and $c$ and are related, respectively, to the position and width of the sigmoidal. Physically this can be interpreted analogously to a transition between two states; alive and dead bacteria. The width of this transition is associated with its cooperativity, that is, the sharper the transition the higher the cooperativity. This does, however, not imply any cooperative peptide–peptide or peptide–bacteria interaction. It is a mere measure of efficiency to inhibit bacterial growth that will be influenced by several stochastic processes, such as cell-to-cell variations of the average number of partitioned peptides.

Mathematically, $F([P]; b, c)$ can be associated to a CDF, that is, it measures the probability of finding a certain number (or fraction) of inhibited cells at a given peptide concentration. This allows to derive the PDF by calculating the derivative, $f([P]; b, c) = \partial F([P]; b, c)/\partial[P] = \partial\phi_{IG}/\partial[P]$, as well as the inverse CDF, $F^{-1}(\phi_{IG}; b, c)$, that is, the $\phi_{IG}$-quantiles. This maps $\phi_{IG}$ values to the inhibitory peptide concentrations $IC_x$, where $x$ is the corresponding inhibited bacterial percentage; by definition, $IC_{99.9} \equiv MIC$. In addition, the set of $b$ and $c$ values as a function of $n_{cell}$ can be interpolated to obtain, for example, a continuous trend of $IC_x$ as a function of $\phi_{IG}$.

Note that, by definition, the derived PDFs express the probability of having a 'killing' event at a given $[P]$.

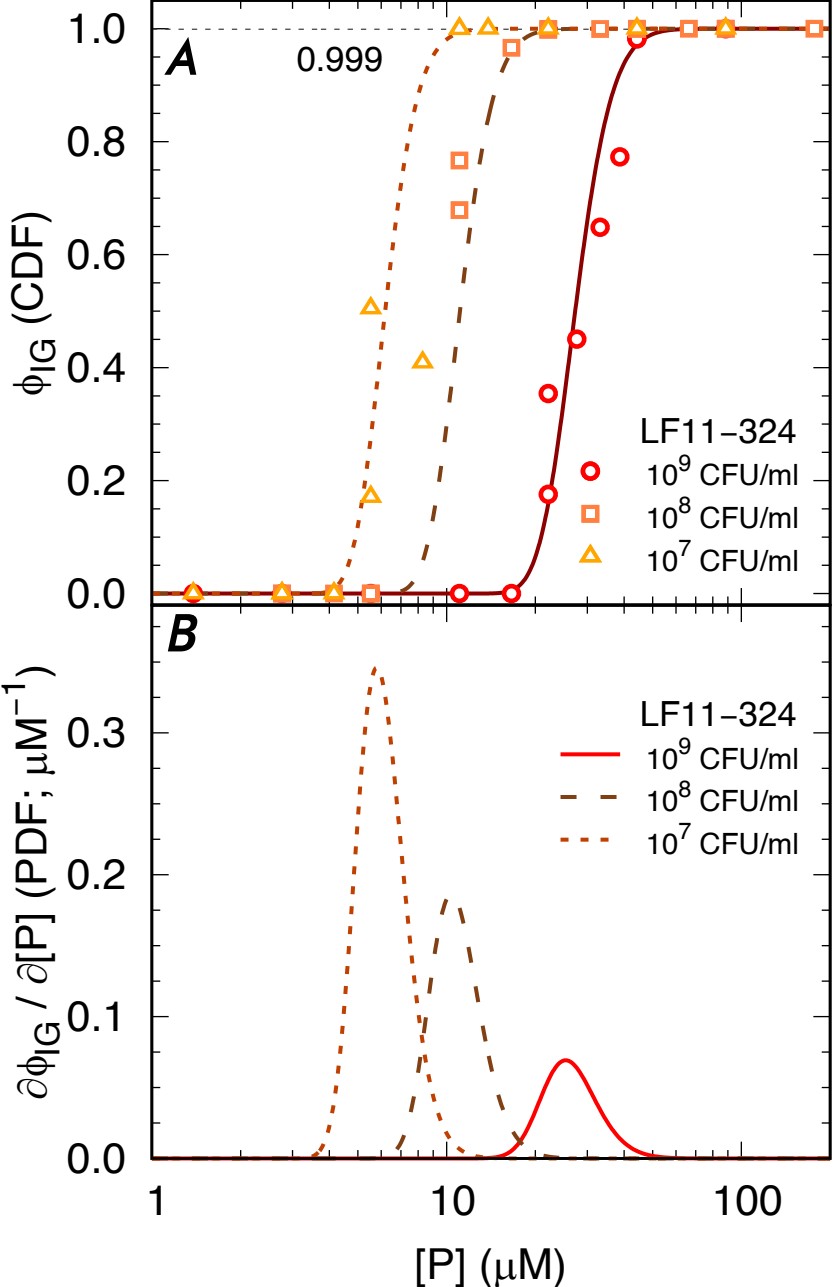

**Appendix 3—figure 1.** Statistical analysis of the AMP-induced bactericidal events. (**A**) Selected $\phi_{IG}$ data for LF11-324 and corresponding fits with the Gompertz function. (**B**) Corresponding probability density functions (PDFs).

## Appendix 4

### Tryptophan fluorescence

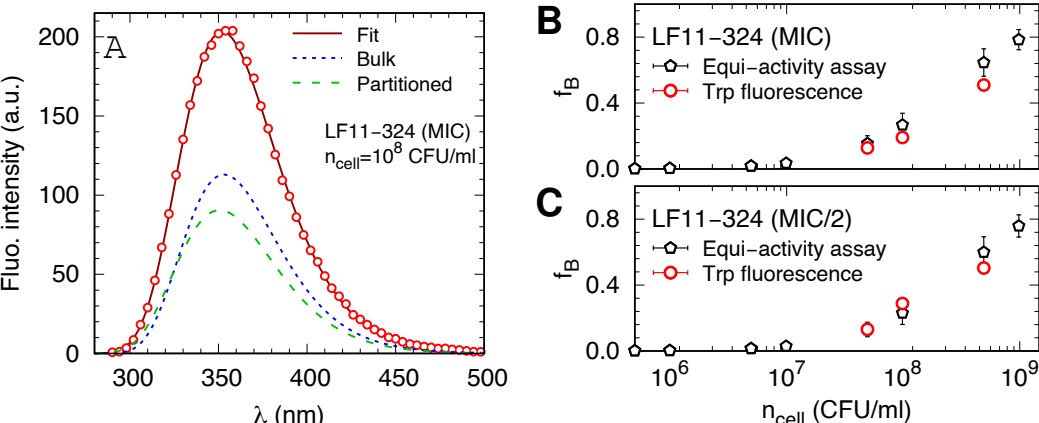

**Appendix 4—figure 1.** AMP-partitioning study in live bacteria based on Trp-fluorescence. (**A**) Example of Trp fluorescence analysis in LF11-324 systems. The solid line is the best fit, and the dotted and dashed lines represent the Trp emissions from peptide in bulk and cell-associated, respectively. Data were fitted with *Equation 4*. (**B**, **C**) Comparison between $f_B$ values obtained from the Trp fluorescence analysis (red dots) and the equi-activity analysis from the susceptibility assay (black pentagons). Error bars in panel (**B** and **C**) are the associated standard deviations of the adjustable parameters obtained from the analysis of the equi-activity assay (black pentagons) and Trp-fluorescence (red dots).

The native fluorescence of the single Trp residue present in LF11 peptides was exploited to validate the partitioning investigation through the equi-activity analysis. For every system, emission signals from partitioned peptides exhibited a weak blue shift, with $\lambda_{max}^{B}$ values in the range 346–354 nm for LF11-215, and 340–350 nm for LF11-324. $\Gamma_B$ values showed no significant variations, instead. Interestingly, these values are consistent with a scenario in which a significant amount of partitioned peptides are heterogeneously dispersed in a polar environment and in a configuration allowing full dynamics of the Trp residues (*Burstein and Emelyanenko, 1996*; *Ladokhin et al., 2000*). This is consistent with the $N_P^{max}/N_B$ values calculated via $\zeta$-potential measurements, suggesting that a relevant portion of partitioned AMPs are still in monomeric state in the intracellular space.

Given that $f_B = n_{cell}N_B/(N_A[P])$, $f_B$ values extracted via Trp fluorescence were compared with those obtained from the antimicrobial activity assays at $[P]$ = MIC and MIC × 0.5 (LF11-215; data not shown). These two independent methods gave comparable $f_B$, confirming the validity of $N_B$ values.

