## [Editor Report]

This article presents groundbreaking data on the effects of a family of antimicrobial peptides on bacterial cells, obtained by time-resolved small-angle X-ray and neutron scattering experiments coupled to stopped-flow mixing. Application of this approach to cells is highly innovative. The main result is that the peptides reach the cytosol in a few seconds, where their accumulation at high concentrations finally kills the bacteria.

---

## [Decision Letter]

**Decision letter after peer review:**

Thank you for submitting your article "Lactoferricins access the cytosol of *Escherichia coli* within few seconds" for consideration by *eLife*. Your article has been reviewed by 3 peer reviewers, and the evaluation has been overseen by a Reviewing Editor and Gisela Storz as the Senior Editor. The following individual involved in review of your submission has agreed to reveal their identity: John Seddon (Reviewer #1).

Essential revisions:

The 3 reviewers have found that your manuscript is potentially important and impactful. They agree that no additional experiments are necessary but that the data analysis and conclusions should be revisited. Here are listed the main changes that are requested in the revision of your manuscript.

1) Data analysis

The changes induced by AMPs on scattering data are not strong. Thus, the analysis of these data, based on a delicate fitting process and 7 adjustable parameters, has to be precisely assessed. I recommend that you address carefully the concerns #1,2, 4-7 of Reviewer 3, and that of Reviewer 1 on this point.

2) Interpretation of the data

The discussion and your conclusions on the bacterial killing mechanism should be also revised by addressing remarks from Reviewer 2 (#1 and 2) and from Reviewer 3 (#8 and 10). They also suggest that you cite more references to other work in your discussion (see Reviewer 2's comments, and Reviewer 3 point #3).

*Reviewer #1 (Recommendations for the authors):*

Some specific comments:

P11, line 309. What is meant by 'momentum'? Is this a misprint?

Lines 313 and 320. The References in the Reference List (p19) should be cited as

Max et al., 2021a, and Max et al., 2021b.

P19, line 652. The Reference Lohner et al., 2008 is incomplete.

P23, line 779. The symbols in B and C are black pentagons, not diamonds.

*Reviewer #2 (Recommendations for the authors):*

1) Mechanism of action

– Several previous studies have shown peptide accumulation in the cytosol [Sochaki 2011, Jepson 2016, Snoussi 2018, Zhu 2019, Savini 2020, Kaji 2021]. Only some of them are cited in the article. All of them should be discussed.

– All previous studies [Jepson 2016, Snoussi 2018, Savini 2020, Kaji 2021], including the direct microscopic observations of Weisshaar and coworkers [Sochaki 2011, Zhu 2019], showed that peptide penetration to the cell cytosol is a consequence of peptide-induced perturbation of permeability of cell membranes. Once the cytosol becomes accessible due to membrane perturbation, association of cationic peptides to intracellular anionic molecules (such as DNA) is an obvious consequence. It is not clear to me if the authors think that in their case peptide accumulation inside the cytosol takes place after perturbation of outer and inner membranes, or without pore formation. In the manuscript, peptide entry in the cytosol is discussed in terms of "translocation", which in general is used to indicate peptide crossing of cell membranes, without pore formation. The authors write "Our experimental setup is not sensitive to directly observe either transient membrane pores or other membrane defects". However, they observed changes in periplasmic and cytosolic SLD. At the end of the introduction, summarizing the study results, the authors write that "collateral damage of the bacterial cell envelope (loss of LPS packing, loss of positional correlations between outer and inner membranes, vesiculation/tubulation, cell shrinkage).… occurred at later time points" with respect to cytosolic entry. However, in Figure 2, this possible difference in time scales is not apparent to me. LPS packing, periplasmic SLD, cell size, all seem to follow kinetics that are rather similar to the cytosolic SLD. Indeed, in the results the authors write "The attack of AMPs first shows up in our time-resolved data by changes of the LPS packing density, as well as periplasmic and cytoplasmic SLDs" and "For LF11-324, the permeabilization of the cytoplasmic membrane occurred as fast as 3- 10 s after mixing". Could the authors clarify these central points? Incidentally, could they comment why the measurements are sensitive to LPS packing, but not to membrane permeability? Don't SANS measurements with different water/deuterated water mixtures help to address this aspect?

– The time for access to the cytosol reported by the authors (less than 3 s) is much faster than previously reported times for other peptides [Sochaki 2011, Zhu 2019]. However, the way this point is discussed in the manuscript seems to imply that this difference is due to the higher time resolution of the technique ("the superior time-resolution and sensitivity.… of synchrotron USAXS/SAXS allowed us to demonstrate.… that AMPs are able to reach the cytosolic compartment of bacteria on the seconds time scale and thus much faster than previously considered"). This sentence might be misleading. The previous studies reported times in the minutes range for peptide access to the cytosol but they did have the resolution needed to observe faster events. Therefore, the different times reported in the various studies are probably due to the specific peptides being studied. Incidentally, in the case of LL-37, bacterial growth stops well before cytosolic entry of the peptide [Sochaki 2011, Zhu 2019]. It is also worth mentioning that in the present study cytosolic entry starts after a few seconds but it does not reach equilibrium even after several minutes (See Figure 2 and Figure 6). If one had to define a lifetime for this process, from Figure 2, I guess that it would be in the order of a couple of minutes. These aspects should be discussed with more caution and detail.

– The authors write "the damage of the cell envelope is a collateral effect of AMP activity that does not kill the bacteria" and "the impairment of the membrane barrier is a necessary but not sufficient condition for microbial killing by lactoferricins". What data support this conclusion? The two effects (membrane perturbation and cytosolic entry) happened on comparable time scales. Bacterial killing was not studied during the SAS experiments. Cell-envelope perturbation took place at sub-MIC concentrations (e.g. 0.7*MIC), but these concentrations are still bactericidal: even if they do not sterilize the sample, a significant fraction of bacteria is killed and would influence the SAS signal. Could the authors assess bacterial growth (in size and number) over the 440 s of kinetic measurements? It is surprising that other (subtler) effects are characterized, and not this macroscopic property. At least, what happens to the optical density of the sample in this timeframe?

– As an argument in favor of interaction with cytosolic targets as the main mechanism of killing, the authors note that the peptide with the lowest MIC reaches also a lower concentration inside the cytosol. They write "Two factors emerge as key components for AMP efficacy: (i) a fast translocation through inner and outer membranes, rapidly reaching extremely high cytosolic AMP concentration levels (100 mM), and (ii) an efficient shut-down of the bacterial metabolism, that is the lower the number of 'needed' AMPs in the cytosol the better". However, the difference in cytosolic concentrations between LF11-324 and LF11-215 is only a factor of 1.4, and the difference in MIC values is similar. However, MICs are determined by twofold dilutions and therefore this difference is probably within the experimental error [Loffredo 2021]. Indeed, at some cell densities the two MICs are comparable. LF11-324 is much faster in reaching the cytosol, than LF11-215, but this property is not reflected in a comparably higher antimicrobial activyt. Finally, a structure/activity correlation is not very solid with just two peptides.

– The authors write "An efficient shut-down of metabolism (is the) primary cause for bacterial killing", and "The primary cause for bactericidal or bacteriostatic activity of the presently studied peptides is thus.… a fast and efficient shut-down of bacterial metabolic activity". What data support this conclusion? Bacterial metabolic activity was not studied.

Is high peptide accumulation in the cytosol consistent with the observed decrease in SLD?

2) Peptide-cell binding

– The intrinsic fluorescence of the peptide was used to follow peptide/cell binding directly, after subtraction of the background due to cell emission. This is a very interesting approach, but was the intensity of this background, compared to the signal?

– The model used by the authors to derive information on peptide/cell interaction from the cell-density dependence of peptide activity is strictly related to that previously proposed by Savini [2017] and Loffredo [2021]. Peptide/cell binding measurements by fluorescent spectroscopy were previously reported by Roversi [2016], Savini [2020] and Kaji [2021]. The fact that the high majority of peptides partition inside the cell is consistent with the recent results by Kaji [2021] and with the binding measurements of Savini [2020].

– Appendix 4 is not sufficiently clear. What do the authors mean exactly by "the probability distributions of inhibiting bacterial growth" and by "cooperativity of killing"? If I understand correctly, in appendix 4 a distribution of "activities" is hypothesized. Is this compatible with the fixed concentration of bound peptides implied in Equation 1?

*Reviewer #3 (Recommendations for the authors):*

1. In the fit model, the authors vary many parameters such as size, periplasmic distance, the width of the periplasmic distance, number of LPS per cell, in addition to scattering length densities (SLDs) of the cytoplasm, periplasm and peptidoglycan respectively. This is 7 parameters in total and in additional 2 more is used for SAXS modeling of additional OM vesicles. Considering the complexity of the structure, this is not unreasonable but freely fitting SLDs without taking into account constraints of mass balance, changes in volume etc. in addition to covariation between parameters might lead to severe pitfalls. I also wonder about the choice considering that there is a lot of potential contributions that are ignored. What about tubules forming from the bilayer, changes in SLDs by peptide insertion into the IM and OM, proteins and flagella. If the peptide translocates there might also be possible structural changes and phase separation within the cytoplasma. Can the authors please comment on this?

2. The changes in the TR-SAXS are rather minor and the main effect seem to be net reduction in the intensity- interestingly this occurs over the whole Q-range and it seems that the bulk of the effect can be described by scaling the whole data set indicating a loss of material through e.g. precipitation. This should be commented on in the discussion of the results. Nevertheless, it also seems there is slight effect of the overall size of the bacteria which can be corroborated with results from TEM. In some cases, like Figure S2 A, some hint of change in the scattering at intermediate Q. These are the length scale where typical changes in the bilayer scattering are also observed upon addition of peptide. This is simply because of the changes in the electron density of (in particular) the hydrocarbon interior of the bilayer in addition to possible membrane thinning (or thickening) etc. How can the authors jump to the conclusion that changes in the LPS surface density and membrane roughness are the only relevant effects to consider? I am not claiming that these are not relevant parameters. However, what is the rationalization behind not considering potentially important changes in contrast and electron density distribution caused directly or indirectly by the peptide that would cause the same effects? The fit curves, at least a few representative examples for each data set should also be shown.

3, Related to 2: The authors simply seem to ignore most of the work done by other groups on small-angle scattering, model membranes and antimicrobial peptides. After all, in this project, that should be a starting point for discussion of the much more complex scattering data from live bacteria.

See e.g., work by:

Meikle et al. Journal of Colloid and Interface Science 2021 587, 90-100.

Nielsen, J. E et al. BBA – Biomembranes 2019, 1861, 1355-1364.

Nielsen, J. E. et al. Soft Matter 2018, 11, 37-14.

Nielsen, J. E. et al. J. Coll. Int. Sci. 2021, 582, 793-802.

Castelletto et al. Langmuir 2012, 28 (31), 11599-11608.

Dehsorkhi e al. Langmuir 2013, 29 (46), 14246-14253.

Narayanan, T et al. J Phys Chem B 2016, 120(44):11484-11491

Qian et al. Biophys. J. 2008, 94 (9), 3512-3522.

Lee, C.-C et al. Biophys. J. 2011, 100 (7), 1688-1696.

Rai, D. K.; Qian, S., Sci. Rep. 2017, 7 (1), 3719.

4. The scattering curve for the peptide Figure S1 B looks very much like a (thin) sheet structure rather than unimeric peptide and a random mass fractal aggregate. The authors should verify this. I suppose a Q-2.4 vs Q-2 dependence would look rather similar in log-log plot.

5. The authors use DMSO as a cosolvents. This mixed with water might give residual background scattering due to partial miscibility at high Q. Also, it seems risky for experiment. May the authors comment?

6. In Table 1 they report different values for neutrons and X-rays. Still, it is claimed that the analysis was done using joint fits of SANS/SAXS which cannot be right then. Can the authors clarify?

7. Figure 1A: Why is the Q-range cut for the data with peptide? Are any aggregates observed at low Q?

8. Figure 2A: the effect on LPS packing seems to be almost independent on concentrations and MIC, considering typical error bars (not shown?). This should be discussed in the context of the proposed mechanism.

9. Figure 3 A: "visibility limit" should rather be "detection limit"

10. I am a bit puzzled by saturation in zeta potential observed below MIC. I don't quite see why this is must indicate translocation as the peptide embedded in the membrane and possible release of ions and larger charged molecules may counteract the effect.

[Editors' note: further revisions were suggested prior to acceptance, as described below.]

Thank you for submitting your article "Lactoferricins access the cytosol of *Escherichia coli* within few seconds" for consideration by *eLife*. Your article has been reviewed by 2 peer reviewers, and the evaluation has been overseen by a Reviewing Editor and Gisela Storz as the Senior Editor. The following individuals involved in review of your submission have agreed to reveal their identity: John Seddon (Reviewer #1).

Essential revisions:

The reviewers consider that your work is important, since it brings new insights on the effects of antimicrobial peptides on bacterial cell using time-resolved scattering data. However, they think the current version, although improved, still tends to over-interpret the data and leads to misleading conclusions. I recommend that you revise once more your manuscript and include the changes listed by referee #2.

*Reviewer #1 (Recommendations for the authors):*

In my view, the authors have considered all of the referees' comments carefully and have gone to considerable lengths to address the points raised. The manuscript has been significantly modified and improved, and I am now happy with it.

*Reviewer #2 (Recommendations for the authors):*

Most of my comments have been addressed in the revised version, which is much clearer. However, thanks to the improved clarity, some critical aspects of the data interpretation in the paper are now even more evident. I am still convinced that the data reported here are very important, but they should be interpreted much more cautiously, as I discuss below. I strongly suggest that the manuscript is reformulated in terms of what was actually measured (i.e. leakage of cell membranes, rather than peptide accumulation in the cytosol)

1) If I understood correctly the authors' replies and corrections, they did not measure peptide accumulation inside the cytosol in the SAS time-resolved experiments. This aspect was not clear to me in the previous version, and I judged rapid peptide accumulation in the cytosol as a "solid finding". However, based on the replies to my queries, important sentences in the abstract and even in the title appear to be misleading. Some examples "Lactoferricins access the cytosol of *Escherichia coli* within few seconds"; "AMPs.… reach the cytosol within less than three seconds".

In the revised version, they write "We additionally checked for contributions of peptides to the SLDs of each bacterial compartment, including inner and outer membranes. The associated changes were found to be insignificant, however". In one of their replies, they write "Scattering of AMPs reaching the cytosol is buried in this signal, i.e. does not significantly contribute to the cytosolic SLD. The observed changes upon the addition of AMP are due to a loss of the low-molecular-weight molecules only"; and "differences in cytosolic and periplasmic SLD…prove permeation and loss of materials, from which we infer and suggest the presence of membrane defects or transient pores"

Therefore, the measured parameters are changes in SLD of periplasm and cytosol, which "need to be due to a leakage of inner and outer membranes". Indeed, the authors write "About cytosolic entry, we can only speculate that it occurs in similar time scales than cytosolic leakage". The phenomenon happening in a few seconds, in a concentration-dependent way, is the leakage of cell membranes.

2) Based on the previous point, conclusions on the mechanism of killing, reported in the abstract, appear to be misleading. I refer to sentences such as the following: "obstruction of physiologically important processes" is the "primary cause for bacterial killing"; "the damage of the cell envelope emerged from our analysis as a collateral effect of AMP activity that does not kill the bacteria. This implies that the impairment of the membrane barrier is a necessary but not sufficient condition for microbial killing". This point is very important, as it is commonly accepted that the mechanism of killing by AMPs is perturbation of the permeability of bacterial membranes, while the abstract implies that this is not the case.

I specifically asked about this point in my previous comments and the authors replied "This conclusion is given by the fact that the cell-envelope damage after one hour of incubation is AMP concentration-independent.… Given that the same macroscopic effects at different AMP concentrations do not correlate to different activities, it seems reasonable to suggest that the damage to the cell envelope is not the primary cause of growth inhibition."

If I understood correctly, here by "cell-envelope damage" they refer to LPS packing, SLD of peptidoglycan layer, periplasm thickness and membrane ruffling. These parameters change in an AMP-concentration independent way. However, these effects are not the commonly accepted mechanism of bacterial killing by AMPs, which is membrane permeabilization. Peptide-induced leakage was actually concentration-dependent, as detected by changes in SLD of periplasm and cytosol: "Differences in cytosolic and periplasmic SLD…prove permeation and loss of materials, from which we infer and suggest the presence of membrane defects or transient pores". Therefore, peptide-induced leakage of the cell membrane cannot be discarded as the cause of bacterial killing. Actually, as discussed above, peptide access to the cytosol is only inferred from cell membrane leakage. The two phenomena cannot be decoupled in the data and therefore in my opinion it is not possible to distinguish between the two as the cause of bacterial killing. Indeed, peptide accumulation in the cytosol could be just a collateral effect of membrane permeability (with internal anionic molecules becoming accessible to the peptide), as several researchers have hypothesized.

3) Comparisons of peptide concentrations and activities in SAS and MIC experiments are questionable.

In replying to one of my questions, the authors specified that the activity was determined by MIC measurements, in a growth medium, which however was not present in the SAS experiments. It is well known that medium components can significantly reduce peptide activity, by sequestration effects. In particular, lactoferricin-derived peptides are very sensitive to growth-medium composition (Journal of Applied Bacteriology 1994, 77, 206-214). Therefore, any comparison regarding active concentrations in the two datasets should be considered very cautiously.

The authors write: "even at growth-inhibited fractions of just 1%, we observed much the same structural changes of the bacterial ultrastructure as at quasi fully growth-inhibited *E. coli*…bacteria are able to recover at sub-MIC concentrations from a severe collateral damage of their cell-envelope. Consequently, this damage cannot be the killing cause for the bacteria". As discussed above, membrane leakage is not included in the "structural changes" considered by the authors, so the above sentences do not apply to peptide-induced membrane permeabilization. Even so, it is very likely that an AMP concentration not inhibiting bacterial growth in a medium becomes much more active when the medium is not present.

Activity could have been measured under the same conditions of the SAS experiments, by determining bacterial killing, rather than growth inhibition.

Other points:

In my previous comments, I asked "If I understand correctly, in appendix 4 a distribution of "activities" is hypothesized. Is this compatible with the fixed concentration of bound peptides implied in Equation 1?". The authors replied: "The number of bound peptides N_B_ needs to be conceived as a thermodynamic average over a distribution of peptides that will insert into the cell". However, cell to cell variations in the number of bound peptides should follow a Poisson distribution. Since the average is in the order of 10^7^, wouldn't fluctuations be negligible?

[Editors' note: further revisions were suggested prior to acceptance, as described below.]

Thank you for resubmitting your work entitled "Lactoferricins access the cytosol of *Escherichia coli* within few seconds" for further consideration by *eLife*. Your revised article has been evaluated by Gisela Storz (Senior Editor) and a Reviewing Editor in consultation with the previous reviewers.

The manuscript has been improved but there are some remaining issues that need to be addressed, as outlined below:

A third round of revision is unusual, but the authors chose to introduce only minimal modifications to the manuscript, even if they agreed that three central conclusions of their paper are derived based on "circumstantial arguments". As such, these statements do not belong in the title, abstract, or Results sections of an *eLife* article, even though they can be presented as hypotheses in the Discussion section. In addition, the indirect arguments presented have some problems. The article is interesting and should be published, but some significant modifications are essential before it can be accepted.

1) Lactoferricins reach "cytosolic concentrations of about 100 mM"

The circumstantial argument goes along these lines:

– from the cell concentration dependence of the peptide MIC, a peptide water-cell partition constant was estimated;

– from this constant, the total number of peptides bound to each cell (under certain conditions) was estimated;

– from zeta potential measurements, using several assumptions, a maximum number of peptides bound to the outer LPS layer was estimated;

– by comparing these numbers, it was estimated that less than 5% of the cell-bound peptides are bound to LPS;

– assuming that binding to other membrane leaflets is comparable to the association to the LPS layer, it was estimated that 92% is in the cytosol.

In addition to its very indirect nature (which should be made explicit in the article), based on several assumptions , the main problem with this line of reasoning is that peptides that are not bound to LPS could be in the inner leaflet of the outer membrane, or in the plasma membrane (for a total of 3 leaflets), but also in the periplasmic space or associated to the peptidoglycan layer. Indeed, in the Results section, the authors correctly use the term "inner compartments". Therefore, the cytosolic concentration cannot be estimated. Incidentally, the calculations and assumptions used to estimate a concentration value of about 100 mM should be clarified.

One possible solution to this issue could be to discuss the data in terms of intracellular or inner compartments (meaning everything inside the outer membrane) concentrations, rather than cytosolic.

Some (non-exhaustive) examples.

Line 21: please substitute "cytosolic" with "intracellular"

Line 86: The final cytosolic->The final intracellular

Line 88: cytosolic-> intracellular

Line 369: "is located within the cell wall"->is associated to the LPS layer"

Line 447 "cytosolic compartment"->"cell interior"

Line 449 "reaching extremely high cytosolic AMP concentration levels"->intracellular

2) "Lactoferricins access the cytosol within a few seconds"

No data are available on the kinetics of the cell entry process. Indeed the authors replied "In the manuscript we do not give any information about the full kinetics of AMP entry". Only membrane leakage was measured. MIC data (on which the above considerations on cytosolic accumulation are based) are measured hours after peptide addition. Therefore, the speed of cell entry should not be mentioned, as it was not measured.

Some (non-exhaustive) examples of the modifications required:

Line 1: the title should be modified to reflect only conclusions for which data are available (e.g. Lactoferricin perturb cell membranes of *Escherichia coli* within a few seconds and accumulate inside the cell).

Line 20: please remove "and reach the cytosol"

Line 22: please substitute "reaching cytoplasm and lowest cytosolic" with "causing leakage and lowest intracellular".

Line 82: Please modify the sentence "Coupling the observed rapid loss of cytoplasmic content with peptide partitioning data we inferred that the studied peptides are able to reach the bacterial cytosol just within few seconds" to "We observed that the studied peptides are able to cause loss of cytoplasmic content just within few seconds".

3) Killing is due to peptide accumulation in the cytosol, rather than to membrane perturbation, which is "a collateral effect of AMP activity that does not kill the bacteria".

In this case, the circumstantial argument is:

– different concentrations of the LF11-324 peptide (at the MIC and sub MIC) have "different leakage kinetics", but eventually "equilibrate to the same cytoplasmic density" (after 1 h, Figure 2B).

– since "the overall degree of leakage is the same for both inhibited-fractions of ~1% and >99.9%" it is "more than reasonable to suggest that permeability alone cannot be the sole cause of bactericidal activity".

– it is hypothesized that the cause for killing must be peptide accumulation in the cytosol (no evidence for this conclusion is provided).

The main problem with this (again indirect) line of reasoning is that also the number of cell-bound LF11-324 peptide molecules is fairly constant over a wide range of growth inhibition values (approximately from 0% to 75%, Figure 5A, where it changes only from 1x10^7^ to 1.5x10^7^). Therefore, based on considerations similar to those reported above, also peptide accumulation in the cytosol should be excluded as the cause of the killing. The cause of killing should be discussed much more cautiously, reflecting the fact that both accumulation and leakage take place at sub-lethal concentrations.

Some examples:

Line 22: please substitute "primary cause for" with "possible factor contributing to"

Line 23: please modify the abstract to reflect the fact that both membrane permeability and intracellular accumulation take place at sub-lethal concentrations.

Line 96: please modify the sentence starting with "The primary cause for bactericidal"

Line 449: please do not mention "efficient shut-down of physiological processes", since no data are available on this aspect. In general, all the concluding remarks should be substantially revised.

---

## [Author Response]

Reviewer #1 (Recommendations for the authors):Some specific comments:P11, line 309. What is meant by 'momentum'? Is this a misprint?

Yes, it is indeed a typo. We meant “amphipathic moment”. We modified the text accordingly.

Lines 313 and 320. The References in the Reference List (p19) should be cited asMax et al., 2021a, and Max et al., 2021b.

Unfortunately, we do not see the typo. The two references (a) and (b) are ranked automatically by the used Latex template to their order in the reference list.

P19, line 652. The Reference Lohner et al., 2008 is incomplete.

We thank the referee to noticing this. The correct reference is now in the text.

P23, line 779. The symbols in B and C are black pentagons, not diamonds.

We modified the text and changed diamonds to pentagons.

Reviewer #2 (Recommendations for the authors):1) Mechanism of action

– Several previous studies have shown peptide accumulation in the cytosol [Sochaki 2011, Jepson 2016, Snoussi 2018, Zhu 2019, Savini 2020, Kaji 2021]. Only some of them are cited in the article. All of them should be discussed.

Given the large number of reports on this aspect it is our humble opinion that Loffredo et al., PNAS 2021 is well summarizing most of these works. Further, we already quoted the reports of Sochaki 2011 and Zhu 2019 in the original manuscript. However, following the suggestion of the referee, we added a reference to Kaji et al. 2021.

– All previous studies [Jepson 2016, Snoussi 2018, Savini 2020, Kaji 2021], including the direct microscopic observations of Weisshaar and coworkers [Sochaki 2011, Zhu 2019], showed that peptide penetration to the cell cytosol is a consequence of peptide-induced perturbation of permeability of cell membranes. Once the cytosol becomes accessible due to membrane perturbation, association of cationic peptides to intracellular anionic molecules (such as DNA) is an obvious consequence. It is not clear to me if the authors think that in their case peptide accumulation inside the cytosol takes place after perturbation of outer and inner membranes, or without pore formation.

Clearly, AMPs need to impair the barrier function of outer and inner membranes to reach the cytosol in agreement with all above mentioned studies. Our X-ray and neutron scattering data are, however, not sensitive to the presence or absence of pores or other forms of local membrane perturbation (note the difference to much less complex lipid-only systems, where effects such as, e.g. membrane thinning can be observed). Hence, we used “translocation” in its very generic sense, without being able to specify any details. Nevertheless, using cytoplasmic membrane mimics, we previously found very weak membrane remodeling activity as compared to other AMPs (31). Moreover, LF11-215 does not cause lipid flip-flop in membrane mimics (30), supporting the minimal “disruption” of the lipid bilayer. To clarify this point we added the following comment:

“We stress that the usage of the term 'translocate' does not imply that AMPs are able to pass through inner and outer membranes without any noticeable effect on membrane structure, such as the transient formation of pores, nor does it exclude their presence. It merely refers to a generic, unspecified uptake of the peptides beyond the resolution of the present experiments. We note, however, that LF11-215 was found to partition into artificial lipid membranes, without noticeable effects on lipid flip-flop (30).”

In the manuscript, peptide entry in the cytosol is discussed in terms of "translocation", which in general is used to indicate peptide crossing of cell membranes, without pore formation. The authors write "Our experimental setup is not sensitive to directly observe either transient membrane pores or other membrane defects". However, they observed changes in periplasmic and cytosolic SLD.

Indeed, we can only directly measure the differences in cytosolic and periplasmic SLD. These parameters prove permeation and loss of materials, from which we infer and suggest the presence of membrane defects or transient pores.

At the end of the introduction, summarizing the study results, the authors write that "collateral damage of the bacterial cell envelope (loss of LPS packing, loss of positional correlations between outer and inner membranes, vesiculation/tubulation, cell shrinkage).… occurred at later time points" with respect to cytosolic entry. However, in Figure 2, this possible difference in time scales is not apparent to me. LPS packing, periplasmic SLD, cell size, all seem to follow kinetics that are rather similar to the cytosolic SLD. Indeed, in the results the authors write "The attack of AMPs first shows up in our time-resolved data by changes of the LPS packing density, as well as periplasmic and cytoplasmic SLDs" and "For LF11-324, the permeabilization of the cytoplasmic membrane occurred as fast as 3- 10 s after mixing". Could the authors clarify these central points?

We agree that the different time scales are not immediately apparent from the data presented in Figure 2 and hence our statement might confuse readers. In particular, AMPs reaching the cytosol leaves a fingerprint in changes of the SLD not only of the cytosol, but also the SLD of the periplasm. It turns out that the periplasm is most sensitive to these initial changes. Here the SLD changes at highest peptide concentration already 3 s after mixing. To clarify this issue, we marked the important time points in Figure 2 and changed the text in the revised manuscript accordingly.

Incidentally, could they comment why the measurements are sensitive to LPS packing, but not to membrane permeability? Don't SANS measurements with different water/deuterated water mixtures help to address this aspect?

The scattering model detailed in Semeraro et al., J. Appl. Cryst 2021 introduced the LPS packing term, which originates from positional correlations between the oligosaccharide cores of these lipids. This accounts for the X-ray scattering shoulder at about q~0.1 nm^-1^ (see also Figure 1A). In the presence of AMP this shoulder disappears, which can be fully accounted for by a loss of the above-mentioned correlations. This effect is in-line with the severe membrane-ruffling as observed by TEM, as well as the formation of OMVs. The formation of transient membrane defects is obscured by scattering from other components of the bacteria. Also note, that peptide pores have been only detected with neutrons using very specific experimental conditions (solid supported single-component membranes at reduced levels of hydration) needed to decrease the systems degrees of freedom, see K. He at al., Biochemistry 34, 15614 (1995).

– The time for access to the cytosol reported by the authors (less than 3 s) is much faster than previously reported times for other peptides [Sochaki 2011, Zhu 2019]. However, the way this point is discussed in the manuscript seems to imply that this difference is due to the higher time resolution of the technique ("the superior time-resolution and sensitivity.… of synchrotron USAXS/SAXS allowed us to demonstrate.… that AMPs are able to reach the cytosolic compartment of bacteria on the seconds time scale and thus much faster than previously considered"). This sentence might be misleading. The previous studies reported times in the minutes range for peptide access to the cytosol but they did have the resolution needed to observe faster events. Therefore, the different times reported in the various studies are probably due to the specific peptides being studied.

We thank the reviewer for this comment, the sentence is misleading. The observed kinetics may indeed be specific to the currently studied AMPs. We modified the sentence to avoid ambiguities.

Incidentally, in the case of LL-37, bacterial growth stops well before cytosolic entry of the peptide [Sochaki 2011, Zhu 2019]. It is also worth mentioning that in the present study cytosolic entry starts after a few seconds but it does not reach equilibrium even after several minutes (See Figure 2 and Figure 6). If one had to define a lifetime for this process, from Figure 2, I guess that it would be in the order of a couple of minutes. These aspects should be discussed with more caution and detail.

We agree that this is a subtle point. Our data does not contain any information about when the amount of cytosolic AMP needed to stop bacterial growth is reached. We only see a flattening of the changes of SLDs in the cytoplasm and periplasm, which are given by AMP induced leakage and are secondary effects of AMP’s entry. We discussed this more carefully in the revised manuscript, stating a clearer differentiation among the onsets of cytosolic leakage. About the lifetime of the leakage process, the half-life (for example) depends on both LF11 type and concentration, ranging from about 1 to >5 minutes.

– The authors write "the damage of the cell envelope is a collateral effect of AMP activity that does not kill the bacteria" and "the impairment of the membrane barrier is a necessary but not sufficient condition for microbial killing by lactoferricins". What data support this conclusion?

This conclusion is given by the fact that the cell-envelope damage after one hour of incubation is AMP concentration-independent (regardless of the kinetics, see Figure 2). For example, at a concentration of 19 µM LF11-324 (corresponding to 0.3xMIC) only 1% of all bacteria are inhibited in their growth. At 43 µM LF11-324 (corresponding to 0.7xMIX) this number increases to 90%. Yet, the observed macroscopic effect on the cell-envelope is indistinguishable. Given that the same macroscopic effects at different AMP concentrations do not correlate to different activities, it seems reasonable to suggest that the damage to the cell envelope is not the primarily cause of growth inhibition. We have clarified this point in the revised manuscript.

The two effects (membrane perturbation and cytosolic entry) happened on comparable time scales.

We respectfully disagree with the reviewer, and we understand that plots and descriptions were probably misleading. Both local membrane perturbation (decrease of LPS packing) and macroscopic cell-envelope damage (increase of Δ_OM_ and σ_OM_, e.g. membrane ruffling) do not happen on comparable time scales. The first effect occurs after about 10 s and the latter after 2-4 minutes, and they are both AMP-concentration-independent. About cytosolic entry, we can only speculate that it occurs in similar time scales than cytosolic leakage. This in turn is AMP-concentration-dependent and happens (for LF11-324, for example) at 3-10 s, 10-20 s and 50-120 s for 1.2x, 0.7x and 0.3xMIC. We tried to clarify it in the revised manuscript and modifying Figure 2.

Bacterial killing was not studied during the SAS experiments. Cell-envelope perturbation took place at sub-MIC concentrations (e.g. 0.7*MIC), but these concentrations are still bactericidal: even if they do not sterilize the sample, a significant fraction of bacteria is killed and would influence the SAS signal. Could the authors assess bacterial growth (in size and number) over the 440 s of kinetic measurements? It is surprising that other (subtler) effects are characterized, and not this macroscopic property. At least, what happens to the optical density of the sample in this timeframe?

About growth, unlike the applied susceptibility assay, no growth medium was added to the bacteria after mixing with AMPs (else all other steps were the same in SAS experiments). Consequently, bacteria are not able to grow during the kinetic experiments. Allowing the bacteria to grow during SAS experiments would impede any detailed analysis by a convolution of diverse structural changes at literally all length scales (growth versus AMP-induced destruction). We clarified this issue in the revised manuscript.

About the influence of either killed or alive cells in the SAS signal, note that both are part of the signal. We clarify this point and its consequences in the text:

“A particularly striking result for the LF11-215/LF11-324 end-states is that the peptide-induced effects are similar and independent of peptide concentration (Figure1, Figure2). That is, even at growth-inhibited fractions of just 1%, we observed similar structural changes of the cell envelope, or e.g. LPS packing, as at quasi fully growth-inhibited *E. coli* (see also Figure4 and Figure4–FigureSupplement1).”

– As an argument in favor of interaction with cytosolic targets as the main mechanism of killing, the authors note that the peptide with the lowest MIC reaches also a lower concentration inside the cytosol. They write "Two factors emerge as key components for AMP efficacy: (i) a fast translocation through inner and outer membranes, rapidly reaching extremely high cytosolic AMP concentration levels (100 mM), and (ii) an efficient shut-down of the bacterial metabolism, i.e., the lower the number of 'needed' AMPs in the cytosol the better". However, the difference in cytosolic concentrations between LF11-324 and LF11-215 is only a factor of 1.4, and the difference in MIC values is similar. However, MICs are determined by twofold dilutions and therefore this difference is probably within the experimental error [Loffredo 2021]. Indeed, at some cell densities the two MICs are comparable. LF11-324 is much faster in reaching the cytosol, than LF11-215, but this property is not reflected in a comparably higher antimicrobial activyt. Finally, a structure/activity correlation is not very solid with just two peptides.

Yes, indeed the ratio of the calculated cytosolic concentrations of peptides (at high cell number density) resemble the ratio of the MICs. This is an expected result, given that at such high bacterial concentrations (10^9^ CFU/ml) most of the peptides are partitioned to the cells. However, we respectfully disagree with the reviewer that the difference in MIC values of LF11-324 and LF11-215 is within experimental uncertainty. We performed fine search of the MIC and partial IC_x_ values that goes beyond the “standard” twofold dilution protocol. The applied methodology, detailed in the Methods and Materials section, follows our recent report (Marx et al., Front Med Technol, 2021) and gives MIC values with <10% uncertainty. Also, it is certainly not our aim to derive a structure/activity correlation from studying just two peptides. We clearly state this now in the conclusion of our paper.

– The authors write "An efficient shut-down of metabolism (is the) primary cause for bacterial killing", and "The primary cause for bactericidal or bacteriostatic activity of the presently studied peptides is thus.… a fast and efficient shut-down of bacterial metabolic activity". What data support this conclusion? Bacterial metabolic activity was not studied.

We fully agree with the reviewer that the bacterial metabolic activity was not studied. Indeed, we cannot disentangle the number of possible interactions with negatively charges macromolecules (Zhou et al., PNAS 2019) and metabolites occurring in the cytosol. This could impair diverse biochemical processes or pathways (e.g protein expression, DNA replication, etc.) We appreciate this comment, and we have modified our corresponding statement detailing the considerations above.

Is high peptide accumulation in the cytosol consistent with the observed decrease in SLD?

As detailed in Semeraro et al. J Appl Cryst 2021, the SLD of the cytosol is dominated by low-molecular weight molecules (ATP, metabolites, etc.) and ions. Scattering of AMPs reaching the cytosol is buried in this signal, i.e. does not significantly contribute to the cytosolic SLD. The observed changes upon the addition of AMP are due to a loss of the low-molecular weight molecules only. We detail this in the revised version of the manuscript.

2) Peptide-cell binding

– The intrinsic fluorescence of the peptide was used to follow peptide/cell binding directly, after subtraction of the background due to cell emission. This is a very interesting approach, but was the intensity of this background, compared to the signal?

Upon excitation at 280 nm, the emission spectra from bare bacterial systems surprisingly showed a single and clean Trp-emission band, peaking at 334-336 nm (FWHM ~65 nm). The peak intensity increased linearly from 4.9 (arb. units) at 10^7^ cells/ml to 122 (arb. units) at 5x10^8^ cells/ml; then turbidity effects had to be accounted for. LF11-215 and LF11-324 showed peaks intensities of about 240 (arb. units) and 140 (arb. units), respectively, in the MIC range, centered at 353-354 nm. All these quantities had errors in the decimal position. These values enabled background-subtracted spectra of good signal-to-noise ratio. We added more details in the Methods and Materials section.

– The model used by the authors to derive information on peptide/cell interaction from the cell-density dependence of peptide activity is strictly related to that previously proposed by Savini [2017] and Loffredo [2021]. Peptide/cell binding measurements by fluorescent spectroscopy were previously reported by Roversi [2016], Savini [2020] and Kaji [2021]. The fact that the high majority of peptides partition inside the cell is consistent with the recent results by Kaji [2021] and with the binding measurements of Savini [2020].

The model proposed in our manuscript is indeed that same reported initially by Savini et al., 2016 (not 2017). The only difference is the definition of the “effective” partitioning coefficient. We modified it slightly to be consistent with the notation used in the original peptide partitioning formalism in liposomes (White et al., Meth Enzymol 1998), to which Savini originally referred to. An analogous method was also applied to the case of liposome/detergents systems by Heerklotz and Seelig, Eur Biophys J 2007. The details of the modelling (including all mentioned references) were described in Marx et al., Front Med Technol 2021. Kaji [2021], Roversi [2014] (not 2016) and Savini [2020] are now quoted in the revised manuscript.

– Appendix 4 is not sufficiently clear. What do the authors mean exactly by "the probability distributions of inhibiting bacterial growth" and by "cooperativity of killing"? If I understand correctly, in appendix 4 a distribution of "activities" is hypothesized. Is this compatible with the fixed concentration of bound peptides implied in Equation 1?

The sigmoidal trend for the increase of growth-inhibited cells with peptide concentration can be transform (upon derivation) into probability density functions (PDF), which describe the likeliness of growth inhibition at a given peptide concentration (and cell density). This is analogous to a transition between two states; here: alive and dead. The width of this transition (or width of the probability function) is associated with its cooperativity, i.e. the sharper the transition the higher the cooperativity. This does, however, not imply any cooperative peptide-peptide or peptide-bacteria interaction. It is a mere measure of efficiency to inhibit bacterial growth. Additionally, these PDFs allow us also to derive the ICx (inhibitory peptide concentration) by calculating the so-called φ_IG_-quantiles.

Equation 1 details the thermodynamics of peptide partitioning into bacterial cells. The number of bound peptides N_B_ needs to conceived as a thermodynamic average over a distribution of peptides that will insert into the cell. As such this is consistent with our statistical analysis of bacterial growth inhibition. We have detailed these issues in the revised manuscript.

Reviewer #3 (Recommendations for the authors):1. In the fit model, the authors vary many parameters such as size, periplasmic distance, the width of the periplasmic distance, number of LPS per cell, in addition to scattering length densities (SLDs) of the cytoplasm, periplasm and peptidoglycan respectively. This is 7 parameters in total and in additional 2 more is used for SAXS modeling of additional OM vesicles. Considering the complexity of the structure, this is not unreasonable but freely fitting SLDs without taking into account constraints of mass balance, changes in volume etc. in addition to covariation between parameters might lead to severe pitfalls. I also wonder about the choice considering that there is a lot of potential contributions that are ignored. What about tubules forming from the bilayer, changes in SLDs by peptide insertion into the IM and OM, proteins and flagella. If the peptide translocates there might also be possible structural changes and phase separation within the cytoplasma. Can the authors please comment on this?

The analysis of our scattering data is constructed in the following way. First, we analyzed the structure of *E. coli* in the absence of AMPs (reported in Semeraro et al., J Appl Cryst, 2021) and presence of AMPs. For initial states we jointly analyzed 11 data sets (10 neutron, 1 X-ray) and for end states, we narrowed it down 6. The joint analysis already sets important constraints to the adjustable parameters. In addition, and as detailed in Semeraro et al., J Appl Cryst, 2021, we consider scattering from all entities of the whole bacterium (considering mass balance, volume distributions etc), some of which do not contribute to the observed scattering signal. These include fimbriae, flagella, or ribosomes, i.e. scattering is not sensitive to structural details of the cytosol and this extends to AMP induced phase separation as observed by TEM. Moreover, several parameters need to be fixed, as we now state clearly in the revised version of our manuscript; but see also Table 2 in Semeraro et al., J. Appl. Cryst, 2021. Tubules were only observed in case of O-LF11-215. However, the presence of O-LF11-215 aggregates obscures their possible detection. In analyzing end-states, we also considered peptide-induced changes of membrane structure. This yielded, however, unphysical results. In order to avoid overparameterization, we therefore remained with the minimum of adjustable parameters needed to fit the data. Importantly, model fitting is not executed using a standard least square procedure, but a Monte Carlo genetic-selection algorithm. This yields for each parameter a probability distribution and allows to derive correlations. Finally, after having established initial and end-state structures, kinetic scattering data (for which we have USAXS/SAXS only) are analyzed in a second step using the structural results as boundaries and further constraining the analysis to follow a smooth transition from initial to end-state. We have updated the manuscript thoroughly to make all these points clearer.

2. The changes in the TR-SAXS are rather minor and the main effect seem to be net reduction in the intensity- interestingly this occurs over the whole Q-range and it seems that the bulk of the effect can be described by scaling the whole data set indicating a loss of material through e.g. precipitation. This should be commented on in the discussion of the results. Nevertheless, it also seems there is slight effect of the overall size of the bacteria which can be corroborated with results from TEM. In some cases, like Figure S2 A, some hint of change in the scattering at intermediate Q. These are the length scale where typical changes in the bilayer scattering are also observed upon addition of peptide. This is simply because of the changes in the electron density of (in particular) the hydrocarbon interior of the bilayer in addition to possible membrane thinning (or thickening) etc. How can the authors jump to the conclusion that changes in the LPS surface density and membrane roughness are the only relevant effects to consider? I am not claiming that these are not relevant parameters. However, what is the rationalization behind not considering potentially important changes in contrast and electron density distribution caused directly or indirectly by the peptide that would cause the same effects? The fit curves, at least a few representative examples for each data set should also be shown.

We respectfully disagree with the reviewer. TR-SAXS patterns at different times are sufficiently distinct for our analysis. For clarification, we added a supplementary figure showing representative curves and including error bars and fitting (see the new Figure 2 —figure supplement 4). This also clearly shows that the changes cannot be attributed a mere intensity scaling. Sedimentation does occur though, but at time scales beyond our kinetic experiments. Previous experiments on the same bacterial strain shows signs of such induced intensity changes after 15-20 minutes. We note this in the revised manuscript.

Regarding bacterial size: The bacterial size in fact changes as observed by a decrease of the minor ellipsoid radius (see Figure 2 C, and I).

Regarding the comment on potential scattering contributions at intermediate q: We agree that such information is accessible in lipid-only systems (e.g. vesicles) interacting with AMPs. The situation in bacteria is vastly different however, where such information is superimposed by contributions from the entire ultrastructure. In fact, we do know from our partitioning analysis that inner and outer membranes are loaded with peptides. However, AMPs do not contribute to the average SLD of each compartment, including the membranes (please refer to the reply to reviewer #2, 1st section, last paragraph).

The conclusions on the decrease of LPS packing are based on constraints form the detailed analysis of initial and end states using multiple SAXS/SANS datasets. Based on TEM data (membrane ruffling) this is expected and serves as the most simple and convincing explanation for the observed changes in scattering intensity. Please note that we do not claim that the effects suggested by the reviewer do not take place. However, USAXS/SAXS cannot detect it and we deliberately refrain from any overparameterization. We clarify this in the revised manuscript.

3, Related to 2: The authors simply seem to ignore most of the work done by other groups on small-angle scattering, model membranes and antimicrobial peptides. After all, in this project, that should be a starting point for discussion of the much more complex scattering data from live bacteria.See e.g., work by:Meikle et al. Journal of Colloid and Interface Science 2021 587, 90-100.Nielsen, J. E et al. BBA – Biomembranes 2019, 1861, 1355-1364.Nielsen, J. E. et al. Soft Matter 2018, 11, 37-14.Nielsen, J. E. et al. J. Coll. Int. Sci. 2021, 582, 793-802.Castelletto et al. Langmuir 2012, 28 (31), 11599-11608.Dehsorkhi e al. Langmuir 2013, 29 (46), 14246-14253.Narayanan, T et al. J Phys Chem B 2016, 120(44):11484-11491Qian et al. Biophys. J. 2008, 94 (9), 3512-3522.Lee, C.-C et al. Biophys. J. 2011, 100 (7), 1688-1696.Rai, D. K.; Qian, S., Sci. Rep. 2017, 7 (1), 3719.

We agree that there is a significant amount of literature on AMP effect on lipid-only mimics SAXS/SANS, some of which (by far not all) are quoted by the reviewer above. In order not to confuse the reader with effects observed in such simple mimics, we decided to quote only those papers, which deal with the here studied peptides. Please note, that we are also not referring to most of our own work on AMPs for the same reason!

4. The scattering curve for the peptide Figure S1 B looks very much like a (thin) sheet structure rather than unimeric peptide and a random mass fractal aggregate. The authors should verify this. I suppose a Q-2.4 vs Q-2 dependence would look rather similar in log-log plot.

We respectfully disagree with the reviewer. A q ^(-2.4)^ decay of scattered intensity shows up much different from a q ^(-2)^ decay on a log-log plot (see updated Figure). Also note that the uncertainty of this decay is about 1% as found by data fitting. In any case, in view of scope of the present manuscript we did not deepen the investigation of the O-LF11-215 aggregates.

5. The authors use DMSO as a cosolvents. This mixed with water might give residual background scattering due to partial miscibility at high Q. Also, it seems risky for experiment. May the authors comment?

We initially shared the same concern of the reviewer. Therefore, we subtracted as background PBS, vol% DMSO and 0.01 vol% acetic acid. In addition, the potential effects on both scattering data and susceptibility/partitioning assay were tested in advance. Effects in neither bacterial scattering pattern nor peptide potency were observed as stated in Methods and Materials/ AMP samples section in the original submission of our manuscript.

6. In Table 1 they report different values for neutrons and X-rays. Still, it is claimed that the analysis was done using joint fits of SANS/SAXS which cannot be right then. Can the authors clarify?

X-rays interact very different with matter than neutrons, leading to much different coherent scattering lengths. Hence, X-ray and neutron SLDs are different even if data has been jointly analyzed (see also Figure 1 – supplement 2 B-D for the SLD changes as a function of D2O). In the case of N_OS_, this originates from a biological variation using a bacterial culture grown at a different time. That is, we allowed for different N_OS_ values when jointly analyzing x-ray and neutron data. Results are however equal within experimental uncertainty. Consequently, this nicely demonstrates reproducibility. We clarify this in the revised manuscript.

7. Figure 1A: Why is the Q-range cut for the data with peptide? Are any aggregates observed at low Q?

SAXS data shown in Figure 1A originate from two different experimental runs. The initial state data was recorded at a 10 times higher sample concentration also explaining the overall differences in intensity scale as detailed now in the revised figure caption. The slightly lower q_min_ in the end state data are due to background subtraction issues that are not trivial in the USAXS range and which become more prevalent for less concentrated samples. No trace of aggregate was ever detected, both with and without peptides.

8. Figure 2A: the effect on LPS packing seems to be almost independent on concentrations and MIC, considering typical error bars (not shown?). This should be discussed in the context of the proposed mechanism.

Error bars are present in each panel of Figure 2. They are not detectable when smaller than the dot size. We propose that the concentration-independent decrease of LPS packing is a consequence of the quick partitioning (or even saturation) of the outer LPS leaflet in <10 seconds. That is, for both LF11-215 and LF11-324, the equilibrium amount of peptides that saturate the LPS leaflets is reached at peptide concentrations of <0.3xMIC. This is in line with zeta-potential measurements that reported a saturation at about 0.2-0.3xMIC. We clarified it in the revised manuscript.

9. Figure 3 A: "visibility limit" should rather be "detection limit"

We modified the plot according to the reviewer suggestion.

10. I am a bit puzzled by saturation in zeta potential observed below MIC. I don't quite see why this is must indicate translocation as the peptide embedded in the membrane and possible release of ions and larger charged molecules may counteract the effect.

The saturation of the zeta-potential with peptide concentration solely indicates the saturation of peptides partitioned in the outer LPS leaflet (as a rough approximation). The translocation was inferred from the whole partitioning analysis, i.e. including bioscreen data. Further, because of the effects mentioned by the reviewer we can only provide an upper boundary of for the number of AMPs per LPS (= 1:3).

[Editors' note: further revisions were suggested prior to acceptance, as described below.]

Essential revisions:The reviewers consider that your work is important, since it brings new insights on the effects of antimicrobial peptides on bacterial cell using time-resolved scattering data. However, they think the current version, although improved, still tends to over-interpret the data and leads to misleading conclusions. I recommend that you revise once more your manuscript and include the changes listed by referee #2.

We would like to thank the editors and reviewers for this second round of constructive comments. We replied to each single point of reviewer #2 and modified a few sentences in the manuscript accordingly.

Reviewer #2 (Recommendations for the authors):Most of my comments have been addressed in the revised version, which is much clearer. However, thanks to the improved clarity, some critical aspects of the data interpretation in the paper are now even more evident. I am still convinced that the data reported here are very important, but they should be interpreted much more cautiously, as I discuss below. I strongly suggest that the manuscript is reformulated in terms of what was actually measured (i.e. leakage of cell membranes, rather than peptide accumulation in the cytosol)

We appreciate the critical reading of the reviewer and tried to further stress and clarify the important aspects of the manuscript. Below we reply to each single recommendation. We also added or modified a few sentences in the main text to help the reader to address the most crucial findings.

1) If I understood correctly the authors' replies and corrections, they did not measure peptide accumulation inside the cytosol in the SAS time-resolved experiments. This aspect was not clear to me in the previous version, and I judged rapid peptide accumulation in the cytosol as a "solid finding". However, based on the replies to my queries, important sentences in the abstract and even in the title appear to be misleading. Some examples "Lactoferricins access the cytosol of *Escherichia coli* within few seconds"; "AMPs.… reach the cytosol within less than three seconds".In the revised version, they write "We additionally checked for contributions of peptides to the SLDs of each bacterial compartment, including inner and outer membranes. The associated changes were found to be insignificant, however". In one of their replies, they write "Scattering of AMPs reaching the cytosol is buried in this signal, i.e. does not significantly contribute to the cytosolic SLD. The observed changes upon the addition of AMP are due to a loss of the low-molecular-weight molecules only"; and "differences in cytosolic and periplasmic SLD…prove permeation and loss of materials, from which we infer and suggest the presence of membrane defects or transient pores"Therefore, the measured parameters are changes in SLD of periplasm and cytosol, which "need to be due to a leakage of inner and outer membranes". Indeed, the authors write "About cytosolic entry, we can only speculate that it occurs in similar time scales than cytosolic leakage". The phenomenon happening in a few seconds, in a concentration-dependent way, is the leakage of cell membranes.

We agree with the reviewer. Indeed, we can directly measure the kinetics of leakage process through the kinetics of the scattering length densities (proportional to the mass densities) of cytoplasmic and periplasmic compartments. In the manuscript we do not give any information about the full kinetics of AMP entry, with the exception of O-LF11-215. In this case we do have an estimation. However, what we can do is distinguishing between periplasm and cytoplasm densities, and this gives a constrain to the timing of perturbation of the cytoplasmic membrane. This is indeed about 3 s for the most effective and concentrated peptide (LF11-324). Our claim of AMPs rapidly reaching the cytoplasm is a circumstantial argument derived from adding our peptide partitioning data into the picture. These data show that the majority of peptide is located inside the bacteria, while only a small fraction remains surface-bound (Figure 5 C). In order to induce loss of cytoplasmic content, AMPs need to reach the cytoplasmic membrane. Including further our previous report on cytoplasmic membrane mimics (Marx et al., Faraday Disc. 2021, doi: 10.1039/D1FD00039J), which showed that lactoferricins readily pass through the bilayers, allows us to conclude that the AMPs rapidly reach the cytosol. The kinetics of O-LF11-215 entry in the cell-volume, along with simple partitioning considerations, strengthen this hypothesis. In order to avoid any confusion, we further clarified this line of argumentation in the revised manuscript.

2) Based on the previous point, conclusions on the mechanism of killing, reported in the abstract, appear to be misleading. I refer to sentences such as the following: "obstruction of physiologically important processes" is the "primary cause for bacterial killing"; "the damage of the cell envelope emerged from our analysis as a collateral effect of AMP activity that does not kill the bacteria. This implies that the impairment of the membrane barrier is a necessary but not sufficient condition for microbial killing". This point is very important, as it is commonly accepted that the mechanism of killing by AMPs is perturbation of the permeability of bacterial membranes, while the abstract implies that this is not the case.I specifically asked about this point in my previous comments and the authors replied "This conclusion is given by the fact that the cell-envelope damage after one hour of incubation is AMP concentration-independent.… Given that the same macroscopic effects at different AMP concentrations do not correlate to different activities, it seems reasonable to suggest that the damage to the cell envelope is not the primary cause of growth inhibition."If I understood correctly, here by "cell-envelope damage" they refer to LPS packing, SLD of peptidoglycan layer, periplasm thickness and membrane ruffling. These parameters change in an AMP-concentration independent way. However, these effects are not the commonly accepted mechanism of bacterial killing by AMPs, which is membrane permeabilization. Peptide-induced leakage was actually concentration-dependent, as detected by changes in SLD of periplasm and cytosol: "Differences in cytosolic and periplasmic SLD…prove permeation and loss of materials, from which we infer and suggest the presence of membrane defects or transient pores". Therefore, peptide-induced leakage of the cell membrane cannot be discarded as the cause of bacterial killing. Actually, as discussed above, peptide access to the cytosol is only inferred from cell membrane leakage. The two phenomena cannot be decoupled in the data and therefore in my opinion it is not possible to distinguish between the two as the cause of bacterial killing. Indeed, peptide accumulation in the cytosol could be just a collateral effect of membrane permeability (with internal anionic molecules becoming accessible to the peptide), as several researchers have hypothesized.

We highly respect previous research in the field and have contributed also ourselves to the commonly accepted picture of AMP mechanism in several previous studies. Yet, our data provide insight from a very different angle that allows us to think beyond this picture (which is mostly based on lipid-only model membrane studies, i.e. missing additional potential targets). Moreover, based on only three (very similar) AMPs we certainly cannot claim a shift of paradigm. At the best our study encourages other groups to pursue similar ideas and add further understanding that will allow to further improve the design of AMPs to combat infectious diseases.

We arrived to this conclusion because, despite the different kinetics, all the end-states of LF11-324 systems at the MIC and sub-MIC (and LF11-215 at the MIC) equilibrate to about the same cytoplasmic density (SLD). It means that the overall degree of leakage is the same for both inhibited-fractions of ~1% and >99.9%. Hence, we find more than reasonable to suggest that permeability alone cannot be sole cause of bactericidal activity. So again, we are using a circumstantial argument to decouple physical impairment of the membrane (leakage) from killing. We added a few sentences to further clarify this reasoning.

3) Comparisons of peptide concentrations and activities in SAS and MIC experiments are questionable.In replying to one of my questions, the authors specified that the activity was determined by MIC measurements, in a growth medium, which however was not present in the SAS experiments. It is well known that medium components can significantly reduce peptide activity, by sequestration effects. In particular, lactoferricin-derived peptides are very sensitive to growth-medium composition (Journal of Applied Bacteriology 1994, 77, 206-214). Therefore, any comparison regarding active concentrations in the two datasets should be considered very cautiously.The authors write: "even at growth-inhibited fractions of just 1%, we observed much the same structural changes of the bacterial ultrastructure as at quasi fully growth-inhibited *E. coli*…bacteria are able to recover at sub-MIC concentrations from a severe collateral damage of their cell-envelope. Consequently, this damage cannot be the killing cause for the bacteria". As discussed above, membrane leakage is not included in the "structural changes" considered by the authors, so the above sentences do not apply to peptide-induced membrane permeabilization. Even so, it is very likely that an AMP concentration not inhibiting bacterial growth in a medium becomes much more active when the medium is not present.Activity could have been measured under the same conditions of the SAS experiments, by determining bacterial killing, rather than growth inhibition.

We respectfully disagree with the reviewer’s assessment of our data. As we further clarified in the methods and materials section, MIC assays were performed incubating bacteria and peptide in PBS buffer, and hence analogously to SAS experiments. The growth medium was added in the MIC assays only after incubation to monitor the bacterial growth and asses AMP efficiency. This is a standard procedure applied by many groups, see also Marx et al., Frontiers in Medical Technology. 2021; doi: 10.3389/fmedt.2021.625975.

Other points:In my previous comments, I asked "If I understand correctly, in appendix 4 a distribution of "activities" is hypothesized. Is this compatible with the fixed concentration of bound peptides implied in Equation 1?". The authors replied: "The number of bound peptides N_B_ needs to be conceived as a thermodynamic average over a distribution of peptides that will insert into the cell". However, cell to cell variations in the number of bound peptides should follow a Poisson distribution. Since the average is in the order of 10^7^, wouldn't fluctuations be negligible?

This is a very interesting point. The modified MIC assay with a fine screening of AMP concentration can actually give a measure of these fluctuations. Indeed, the asymmetric sigmoidal trend of growth inhibited-*E. coli* fraction as a function of AMP concentration is very likely the integral of a Poisson (or Poisson-like) distribution, i.e. the CDF. The partitioning analysis (either our proposed version or the one reported firstly in Savini et al. 2016) aims to associate an average N_B_ to each inhibited-fraction value along the above-mentioned distribution.

[Editors' note: further revisions were suggested prior to acceptance, as described below.]

The manuscript has been improved but there are some remaining issues that need to be addressed, as outlined below:A third round of revision is unusual, but the authors chose to introduce only minimal modifications to the manuscript, even if they agreed that three central conclusions of their paper are derived based on "circumstantial arguments". As such, these statements do not belong in the title, abstract, or Results sections of an eLife article, even though they can be presented as hypotheses in the Discussion section. In addition, the indirect arguments presented have some problems. The article is interesting and should be published, but some significant modifications are essential before it can be accepted.1) Lactoferricins reach "cytosolic concentrations of about 100 mM"The circumstantial argument goes along these lines:- from the cell concentration dependence of the peptide MIC, a peptide water-cell partition constant was estimated;- from this constant, the total number of peptides bound to each cell (under certain conditions) was estimated;

We respectfully disagree with the reviewer. Throughout the manuscript we carefully discriminate between estimated, calculated and measured quantities. The number of peptides partitioned in each cell was calculated from robust modeling of the growth inhibition assay. As control, the number of peptides partitioned into each cell was also directly measured via tryptophan spectroscopy (Appendix 4), fully supporting the growth inhibition analysis. In the manuscript we use the term ‘estimated’ because of the degree of variability given by the experimental setups. In order to avoid any misunderstanding with the meaning of estimation we rephrased several sentences in the manuscript accordingly.

- from zeta potential measurements, using several assumptions, a maximum number of peptides bound to the outer LPS layer was estimated;- by comparing these numbers, it was estimated that less than 5% of the cell-bound peptides are bound to LPS;

Also in this case, the *maximum* number of peptides partitioned in the LPS layer was not estimated, but calculated from robust mathematical and physical principles as detailed in Marx et al., Front. Med. Technol. 2021. It must be stressed that an estimation of the exact number of peptides in the LPS layer is not accessible via z-potential, given the complexity of the cell membrane. However, this does not exclude a safe calculation of the maximum number as the highest possible value.

- assuming that binding to other membrane leaflets is comparable to the association to the LPS layer, it was estimated that 92% is in the cytosol.In addition to its very indirect nature (which should be made explicit in the article), based on several assumptions , the main problem with this line of reasoning is that peptides that are not bound to LPS could be in the inner leaflet of the outer membrane, or in the plasma membrane (for a total of 3 leaflets), but also in the periplasmic space or associated to the peptidoglycan layer. Indeed, in the Results section, the authors correctly use the term "inner compartments".

As already stated above, we are unambiguously able to calculate, based on our data, the highest number of AMP that the outer leaflet of the outer membrane can possibly host. This calculation reveals that not more than about 2% of the total amount of partitioned peptides per cell will be found in this leaflet (i.e. most likely less). Note that lipid-only membranes would be unstable at 4-5 AMPs per lipid (see, e.g., Melo et al., Nat. Rev. Microbiol. 2009; or Marx et al., Front. Med. Tech 2021 for LF11 peptides). Multiplying our upper bound value of 2% with the total number of membrane leaflets (4) thus gives us a highly conservative estimate for the number of AMPs that can be possibly associated with inner and outer membranes. It follows that even if we associate as many as possible AMPs with either membrane that still the vast majority of AMPs (92%) will not be located in membranes.

In any case, we agree that we cannot differentiate between cytoplasm and periplasm’s local concentration of peptides (the peptidoglycan is within the periplasmic space). We modified the manuscript as suggested.

Therefore, the cytosolic concentration cannot be estimated. Incidentally, the calculations and assumptions used to estimate a concentration value of about 100 mM should be clarified.

The intracellular peptide concentration is simply given by the size (and henceforth volume) of the cytoplasmic and periplasmic spaces as determined by USAXS/SAXS/DLS and the above described 92% of the total number of cell partitioned peptides. We clarify this in the revised manuscript.

One possible solution to this issue could be to discuss the data in terms of intracellular or inner compartments (meaning everything inside the outer membrane) concentrations, rather than cytosolic.Some (non-exhaustive) examples.Line 21: please substitute "cytosolic" with "intracellular"

Done

Line 86: The final cytosolic->The final intracellular

Done

Line 88: cytosolic-> intracellular

Done

Line 369: "is located within the cell wall"->is associated to the LPS layer"

We modified it with the more appropriate “is located within the membranes”.

Line 447 "cytosolic compartment"->"cell interior"

We modified it with the more appropriate “cytosolic membrane”.

Line 449 "reaching extremely high cytosolic AMP concentration levels"->intracellular

The paragraph is now partially rephrased, the new equivalent sentence is “and subsequently accumulate at high concentrations (80 – 100 mM) in all bacterial compartments”.

2) "Lactoferricins access the cytosol within a few seconds"No data are available on the kinetics of the cell entry process. Indeed the authors replied "In the manuscript we do not give any information about the full kinetics of AMP entry". Only membrane leakage was measured. MIC data (on which the above considerations on cytosolic accumulation are based) are measured hours after peptide addition. Therefore, the speed of cell entry should not be mentioned, as it was not measured.Some (non-exhaustive) examples of the modifications required:Line 1: the title should be modified to reflect only conclusions for which data are available (e.g. Lactoferricin perturb cell membranes of *Escherichia coli* within a few seconds and accumulate inside the cell).

We have difficulties to understand the physical basis of how peptides (given all the derived numbers of cell partitioned peptides and the number of potential targets within the cytosol) can induce cytosolic leakage without reaching the cytosol. However, in order to cut this discussion short, we agree to change the title with a slight modification of the suggested wording “Lactoferricins impair the cytosolic membrane of *Escherichia coli* within a few seconds and accumulate inside the cell”. Yet, our favorite title still is the original and we would be happy to change back upon the word of the editors.

Line 20: please remove "and reach the cytosol"

Done

Line 22: please substitute "reaching cytoplasm and lowest cytosolic" with "causing leakage and lowest intracellular".

We changed to “…speed of permeabilizing membranes and lowest intracellular peptide concentration”

Line 82: Please modify the sentence "Coupling the observed rapid loss of cytoplasmic content with peptide partitioning data we inferred that the studied peptides are able to reach the bacterial cytosol just within few seconds" to "We observed that the studied peptides are able to cause loss of cytoplasmic content just within few seconds".

The proposed sentence misses the gist of our results. We suggest the following rewording/extension: “We observed that the studied peptides are able to cause a loss of cytoplasmic content just within few seconds. Coupling this finding to the derived final (after one hour) intracellular peptide concentrations at full bacterial growth inhibition (~ 100 mM) suggests a rapid peptide uptake on similar time scales and hence, much faster than previously reported…”

3) Killing is due to peptide accumulation in the cytosol, rather than to membrane perturbation, which is "a collateral effect of AMP activity that does not kill the bacteria".In this case, the circumstantial argument is:-different concentrations of the LF11-324 peptide (at the MIC and sub MIC) have "different leakage kinetics", but eventually "equilibrate to the same cytoplasmic density" (after 1 h, Figure 2B).-since "the overall degree of leakage is the same for both inhibited-fractions of ~1% and >99.9%" it is "more than reasonable to suggest that permeability alone cannot be the sole cause of bactericidal activity".-it is hypothesized that the cause for killing must be peptide accumulation in the cytosol (no evidence for this conclusion is provided).The main problem with this (again indirect) line of reasoning is that also the number of cell-bound LF11-324 peptide molecules is fairly constant over a wide range of growth inhibition values (approximately from 0% to 75%, Figure 5A, where it changes only from 1x10^7^ to 1.5x10^7^). Therefore, based on considerations similar to those reported above, also peptide accumulation in the cytosol should be excluded as the cause of the killing. The cause of killing should be discussed much more cautiously, reflecting the fact that both accumulation and leakage take place at sub-lethal concentrations.

We detect some misconceptions. The number of cell bound LF11-324 *monotonously increases* even if it shows a “fairly” constant region around 50%. This is simply due to the fact that the largest fraction of cells “dies” in this AMP concentration range. Figure 4-supp1 clearly shows this fact, or explicitly: The concentration needed to inhibit 1% – ~ 80% of the bacteria varies only within a few µM, but starts to increase rapidly upon approaching the MIC (see also Appendix 3).The trend of N_B_ as a function of inhibited fraction (Figure 5) mirrors this behavior. We all agree that our data provides unambiguous evidence for the peptides being mostly located within the cytosol and periplasm. If the leakage of cellular content and the damage of the cell wall do not suffice to kill bacteria (because the ensemble average of these effects is unambiguously equal for 1% and 99.9% of growth inhibited bacteria – see USAXS/SAXS) there is not much of a choice left than attributing ‘the killing’ to an accumulation of peptides within the bacteria. The most vital molecules for bacterial metabolism, reproduction, etc. are contained within the cytosol, many of which are prone to be highly attractive targets for AMPs (through unspecific intermolecular forces). Besides, the volume of the cytosol exceeds that of the periplasm significantly. Consequently, the majority of the > 92% of intracellular peptides will be located within the cytosol. So in many ways, although not shown directly, killing will be taking place within the cytosol. There are many peptides that are able to cross the membranes without causing severe membrane damage to reach intracellular targets, see, e.g. Indolicin (Subbalakshmi and Sitaram, FEMS Microbiol. Letters 1998). This is only one among the examples cited in the review of Le et al., Antimicrobial Agents and Chemotherapy 2017 (newly added to the MS). We also wish to quote a statement of these authors, which appears to perfectly fit this discussion. “Based on the findings collected in this paper and elsewhere, we should now understand that a substantial number of AMPs possess multifunctional activities at membranal and/or intracellular sites to achieve efficient killing. Furthermore, the intracellular inhibitory mechanisms of AMPs serve equally important roles, contributing to the overall antibacterial efficiency. Unfortunately, these aspects are commonly overlooked or unreported”.

We carefully discuss these delicate points in the revised version of the manuscript.

Some examples:Line 22: please substitute "primary cause for" with "possible factor contributing to"

As discussed in detail above, the accumulation of AMPs in the cell emerges indeed as the primary cause for killing (although we are not able to detail beyond that). Consequently we keep the sentence as is.

Line 23: please modify the abstract to reflect the fact that both membrane permeability and intracellular accumulation take place at sub-lethal concentrations.

This was already stated in the previous revision (line 24): “On the other hand, damage of the cell envelope and leakage occurred also at sublethal peptide concentrations”.

Line 96: please modify the sentence starting with "The primary cause for bactericidal"

We keep this statement as outlined above.

Line 449: please do not mention "efficient shut-down of physiological processes", since no data are available on this aspect. In general, all the concluding remarks should be substantially revised.

We substantially revised several sections of the manuscript including in particular the concluding paragraph and Figure 6. Throughout the discussion we clearly state (because there is no explicit data) that this is a proposal and not further clarified. Given the amount of solid evidence collected within this study we believe that this is a valid suggestion and a highly plausible scenario. Moreover, “physiological process” is the most generic and unspecific wording we are able to come up with. Anything less than that would force us to step back from the significant and unambiguous results from our study.